# Unifying Diverse Decision-Making Scenarios with Learned Discrete Actions

## Abstract

Designing effective action spaces for complex environments is a fundamental and challenging problem in reinforcement learning (RL). Although various action shaping and representation learning methods have been proposed to address some specific action spaces and decision-making requirements (e.g. action constraints), these methods often are typically customized to fixed scenarios and require extensive domain knowledge. In this paper, we introduce a general framework that can apply any common RL algorithms to a class of discrete latent actions learned from data. This framework unifies a wide range of action spaces, including those with continuous, hybrid, or constrained actions. Specifically, we propose a novel algorithm, the *General Action Discretization Model (GADM)*, that can adaptively discretize raw actions to construct unified and compact latent action spaces. Moreover, *GADM* also predicts confidence scores of different latent actions, which can help mitigate the instability of parallel optimization in online RL settings, and serve as an implicit constraint for offline RL cases. Quantitative experiments and visualization results demonstrate that our proposed framework can match or outperform various approaches specifically designed for different environments.

## 1 Introduction

Recent advances in Reinforcement Learning (RL) have yielded many promising research achievements (Vinyals et al., 2019; Berner et al., 2019; Ouyang et al., 2022). However, the complexity of action spaces still prevent us from directly utilizing advanced RL algorithms to real-world scenarios, such as high-dimensional continuous control in robot manipulation (Lillicrap et al., 2016) and structured hybrid action decision-making in strategy games (Kanervisto et al., 2022). These issues lead to extensive challenges in designs of policy optimization (Xiong et al., 2018) and efficiency of exploration (Seyde et al., 2021). In addition, some new areas like offline RL (Fujimoto et al., 2019) and safe RL (Liu et al., 2023) are solving training stability and policy behaviour legitimacy problems under special action constraints, which brings new requirements to the design of action spaces.

To handle these issues, some existing work first elaborately design particular RL methods in raw action spaces. Specifically, deterministic policy gradient methods (Lillicrap et al., 2016; Fujimoto et al., 2018) are designed to handle continuous control problems. Xiong et al. (2018) and Fan et al. (2019) propose some techniques to model the intra-relationship within hybrid actions. Besides, Fujimoto et al. (2019) and Yang et al. (2021) utilize the property whether candidate actions belong to pre-collected offline datasets to suppress the over-estimation problem about Q value. Correspondingly, another idea is to make raw action spaces more suitable for RL training. Action space shaping (Kanervisto et al., 2020) is a classic way to tackle these problems. Particularly, many RL applications in games (Kanervisto et al., 2022; Wei et al., 2022) design specific action discretization mechanisms to simplify the decision-making spaces, leading to the promising performance improvement. Meanwhile, some works propose to learn an action model that abstracts raw actions into latent actions to boost RL training. HyAR (Li et al., 2021) designs a special training scheme with VAE (Kingma & Welling, 2014) to map the original hybrid action space to a continuous latent action space. Some other methods (Shafiullah et al., 2022; Jiang et al., 2022; Dadashi et al., 2022) build prior sets of discrete actions from pre-collected demonstrations, and then design new RL agents on these fixed actions. However, these designs are tailored to specific environments, rendering them unsuitable for diverse decision-making scenarios that encompass arbitrary action spaces. Moreover, for a specific environment, researchers need to invest considerable time and effort in learning related domain knowledge to

preserve the necessity and minimize the redundancy of actions. Furthermore, previous studies on action representation learning have primarily focused on offline RL, with fewer instances in the more commonly used online RL.

In order to address above challenges, we first investigate two important aspects of learning the general action representation: As illustrated in Figure 1, we revisit the potential redundancy of raw action spaces, including semantically repeated discrete action choices in some specific states, unnecessary fine-grained continuous control, and invalid intra-action relationships in hybrid action space. By reducing these invalid, useless and semantically similar actions and remapping the raw action space into a more efficient latent representation, RL agents can focus more on the necessary subset of actions, enhancing both exploration and exploitation (Chandak et al., 2019). In addition, offline datasets can be considered as important priors for action representation learning. For offline RL settings, the well-trained action model can become an implicit constraint for action selection (Fujimoto et al., 2019), thereby effectively alleviating the over-estimation issues in temporal difference (TD) learning. For online RL, these datasets can be used to pretrain the action model, providing an advantageous starting point for subsequent training.

Drawing on these insights, we propose a general framework that partitions the decision-making problem into two components: action representation learning and discrete reinforcement learning, as illustrated in Figure 2. A key element of this framework is our novel *General Action Discretization Model* (*GADM*), which includes a diversity-aware codetable, a state-dependent action encoder and decoder, along with a latent action confidence predictor. Because of its simplicity and scalability, GADM effectively reduces action redundancy and accommodates both offline and online training regimes. For the offline case, we first train *GADM* on the offline dataset, then apply any standard discrete RL algorithm like DQN (Mnih et al., 2015) (model-free) or MuZero (Schrittwieser et al., 2019) (model-based) to the learned discrete action space. For the online case, due to continual online interactions and dynamically changing data distributions, it is necessary to train the action model and RL models together. However, given that the quantity of latent actions remains fixed, while the effective actions fluctuate considerably across different states, it's inevitable that some latent actions become ineffective in certain scenarios. This situation gives rise to a new issue termed as the *latent action out-of-distribution* problem (Figure 3). To mitigate this, *GADM* generates confidence scores for latent actions, wherein higher scores reflect more reliable actions. These scores can be converted into a *latent action mask*, which can then be incorporated into Temporal-Difference (TD) learning and policy gradient as supplementary guidance. Finally, we assess the effectiveness of this framework in both online and offline RL environments, spanning various action spaces such as Gym Hybrid (thomashirtz, 2021), HardMove from HyAR Li et al. (2021), GoBigger (Zhang, 2021), MuJoCo Todorov et al. (2012), and D4RL (Fu et al., 2020a). Results show that *GADM* with naive DQN outperforms previous algorithms specifically designed for corresponding action spaces in both efficiency and performance. Further validation is provided through a series of ablation study experiments and visualizations.

To summarize, the core contributions of this paper can be summarized as follows:

- We propose a novel representation algorithm *GADM* that can learn unified and compact discrete latent actions for different environments with continuous or hybrid action spaces.

- Our proposed framework, *GADM* with arbitrary common discrete reinforcement learning algorithms, is the first decoupling paradigm capable of both online and offline RL training.

- Through quantitative experiments and visualizations, we demonstrate the scalability and efficiency of our method, showing its potential as a foundational design for general decision-making models.

## 2 BACKGROUND

**Markov Decision Process**  In RL, we model a decision-making problem as a Markov Decision Process (MDP) $\mathcal{M}=(\mathcal{S}, \mathcal{A}, \mathcal{P}, \mathcal{R}, \gamma, \rho_0)$, where $\mathcal{S}$ and $\mathcal{A}$ represent the state space and the action space. And the transition function $\mathcal{P}$ assigns to each state-action pair $(s, a) \in \mathcal{S} \times \mathcal{A}$ a probability measure over $\mathcal{S}$, which we shall denote by $\mathcal{P}(\cdot \mid s, a)$; the expected reward function $\mathcal{R}$ assigns to each state-action pair $(s, a) \in \mathcal{S} \times \mathcal{A}$ a probability measure over $\mathbb{R}$, which we shall denote by $\mathcal{R}(\cdot \mid s, a)$. $\gamma \in [0, 1)$ is the discounted factor, and $\rho_0$ is the initial state distribution. A policy $\pi$ is a mapping from states to distributions over the action space. The objective of RL is to learn an optimal policy $\pi$ to maximize the expected discounted return $J(\pi) = \mathbb{E}_{\pi, \rho_0, \mathcal{P}, \mathcal{R}}[\sum_{t=0}^{\infty} \gamma^t r_t]$, where the expectation is taken with respect to the trajectory distribution induced by $\pi$ and environment dynamics.

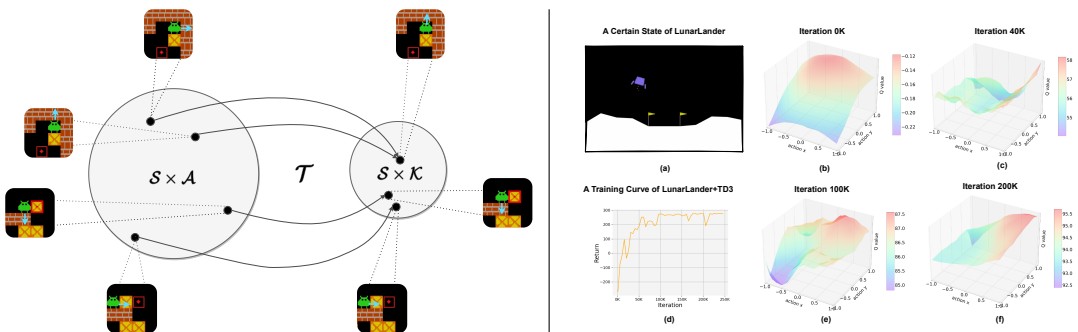

**Figure 1 Left**: Example of a reduction of an MDP's state-action space under an transformation operator $\mathcal{T}$. Actions with the same impact in raw MDP will be mapped to the same latent action. The agent pushing up boxes has similar effect with right push so they have the same latent semantics. The other two state-action pairs would transfer into close locations therefore they have more close latent semantics. **Right**: Figure (a) shows a snapshot of LunarLander, and Figures (b, c, e, f) show the Q value estimated by the neural network in different training stages of the TD3 algorithm in this state. It can be seen that the curved surfaces at different training stages are relatively smooth (i.e. the possible action space redundancy), and the difference between each specific action and nearby actions is not very large. Figure (d) shows the performance changes of TD3 with the iterative training.

**Hybrid Action Space and Action Transformed MDP**   (Masson et al., 2016) presents the idea of a parameterized action space, characterized by a hierarchical structure with two layers. Specifically, $\mathcal{A}_d$ symbolizes a finite collection of discrete actions, while $\mathcal{X}_a$ represents the related continuous parameter space for each action $a \in \mathcal{A}_d$. The action selection process initiates with the choice of a discrete action $a$ from $\mathcal{A}_d$, subsequently followed by the selection of a parameter $x$ from the corresponding space $\mathcal{X}_a$. In order to accommodate a broader hybrid action space as seen in complex video games like StarCraft II (Vinyals et al., 2017), Dota 2 (OpenAI, 2018) and many real-world tasks, we expand this parameterized action space into a general N-layered hybrid action space. To facilitate a more comprehensive understanding of the MDP associated with both the original action space and the discretized action space, we present a detailed math definition in the appendix A.1.

## 3   GENERAL ACTION DISCRETIZATION MODEL

We begin this section by introducing a detailed overview of the entire framework and its corresponding training pipeline in Section 3.1. Subsequently, we dissect the specific components of the General Action Discretization Model (*GADM*) in Section 3.2, including network structures and loss functions. Based on the entire framework, we describe practical algorithms (e.g. *GADM+DQN*) in Section 3.2. We also discuss some underpinning motivations and insights that inspired *GADM* to Appendix A.2.

### 3.1   FRAMEWORK OVERVIEW

**Framework.**   Motivated by the analysis about decoupling action representation learning and reducing action space redundancy (Figure 1 and Appendix A.2), we propose a unified framework leveraging standard RL on a learned discrete action space, designed to accommodate diverse decision-making scenarios with intricate action spaces (refer to Figure 2). The framework operates as a *meta-algorithm*, providing a universal solution for various decision-making problems. The significant challenge of crafting suitable action spaces is resolved through a *two-part decoupled design*: a *representation learning component GADM*, which maps the raw action space to a new discrete action space, and a *discrete RL component* built on these learned discrete action representations. Significantly, this design incorporates two types of models, each with their unique optimization objectives—the action model and the RL model. Under this framework, both action representation learning and RL methods based on discrete action spaces can be optimized in a relatively decoupled manner. This greatly simplifies the previous process of handling complex action spaces (Bester et al., 2019; Fan et al., 2019).

**Dual training pipelines for GADM: offline and online RL.**   The training pipeline for our action models accommodates two distinct scenarios, those being the *GADM for offline RL* and the *GADM for online RL* settings. In offline RL setting, as delineated in Algorithm 2, the action model is initially trained on offline datasets, and then utilized for RL training. However, when directly applied to tasks, this approach may encounter a unique issue referred to as the *pathological latent action space*, characterized by redundant and shifted latent actions. We explore this issue in depth in Section 3.2.

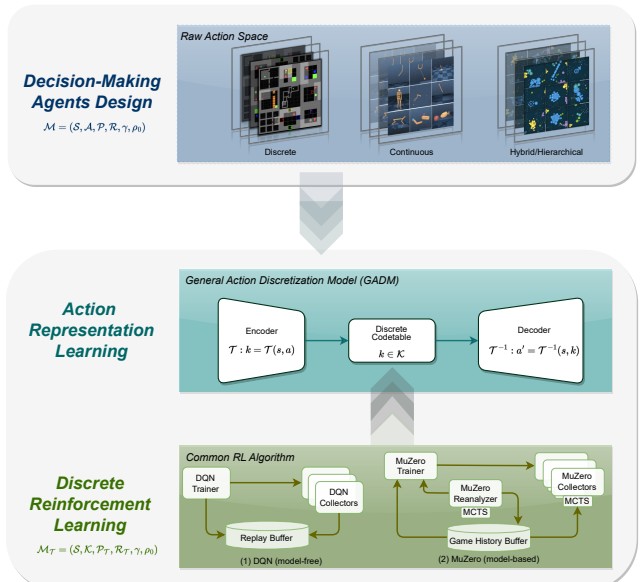

**Figure 2** The unified decision-making framework for various action spaces. It is divided into a *two-part decoupled design*: first, an *action representation learning component, GADM*, followed by a *discrete RL component* that is built upon these learned discrete action representations.

As illustrated in Algorithm 1, the *GADM for online RL* setting facilitates concurrent training of the action and RL models. In this scenario, during RL training, the agent selects a *discrete latent action* $k_t$ based on the current state $s_t$. This latent action is decoded into a *raw action* $a_t$ by the action model. It is then possibly integrated with some exploration mechanisms (refer to A.5.3) before being applied to the environment. The ensuing reward $r_t$ and subsequent state $s_{t+1}$ form a transition sequence $\{s_t, k_t, a_t, r_t, d_t, s_{t+1}\}$, which is stored in the replay buffer. However, this online approach could face unique challenges. With the simultaneous optimization of the *GADM* and RL models, transitions in the replay buffer, gathered across various training iterations via the older action model, could lead to the same latent action mapping to different raw actions.

To tackle this, we introduce a *latent action remapping* technique, akin to the reanalyze operation in MuZero Schrittwieser et al. (2019). In this process, the latent action, determined by the older version of *GADM* in the collected mini-batch $\{s_t, a_t, k_t^{old}, r_t, d_t, s_{t+1}\}$, is remapped to the corresponding latent action using the current action encoder $e_\phi$: $k_t^{new} = e_\phi(s_t, a_t)$. The RL training is then executed on the remapped samples $\{s_t, sg[k_t^{new}], r_t, s_{t+1}\}$ (where $sg$ denotes the stop gradient operation). Additionally, the *GADM for online RL* setting's training pipeline can benefit from an *optional warm-up phase*. During the warmup stage, data can accumulated via a random or expert policy, or from a pre-collected dataset. This data is used to train the action model, providing a solid foundation for the subsequent online stage, as shown in Section 4.1.3.

## 3.2 GENERAL ACTION DISCRETIZATION MODEL

In this section, we delve into the fundamental design of *GADM*, a modified discrete autoencoder (van den Oord et al., 2017) comprising several key components: a diversity-aware codetable, a state-conditioned action encoder and decoder, and a latent action confidence predictor.

**Diversity-aware codetable.** Codetable maintains $K$ latent action candidates $\{z^k\}_{k=0}^{K-1}$. Each of them is a vector of length $N$. We first tried to update the codetable using the exponential moving average trick proposed in VQ-VAE, but we found that different vectors gradually were updated to similar values so that most of them were homogeneous, which limited the number of available actions for RL. To solve it, we design a diveristy-aware codetable, which is initialized by a series of one-hot vectors or bisection points and remains fixed through the entire training process.

**State-conditioned action encoder and decoder.** The standalone values of actions, whether categorical, continuous or hybrid, are inadequate for comprehensive action representation learning. To overcome this, we integrate states as an additional condition in both the encoder and decoder. One significant benefit of state-conditioning is its ability to substantially reduce the number of required

embeddings, or the shape of latent action, in the embedding table. The mathematical definitions for the state-conditioned encoder and decoder are presented below (for detailed network structures, please refer to Appendix A.4.2):

$$\text{Encoder}(e_\phi) : z^e = e_\phi(s_t, a_t), \text{Decoder}(d_\psi) : \hat{a}_t = d_\psi(s_t, z^k), k = \text{argmin}_j \left\| z^e - z^j \right\|_2 \quad (1)$$

In this framework, $\phi$ and $\psi$ represent the network parameters for the encoder and decoder, respectively. The encoder's output embedding is matched in the codetable using the nearest neighbour rule, with the index of the located vector serving as the corresponding *latent action* $k$. We propose the following objective function for joint learning of the encoder and decoder:

$$\mathcal{L} = w(s_t, a_t)\mathcal{L}_{rec}(\hat{a}_t, a_t) + \lambda \left\| z^e - \text{sg}\left[e_\phi(x)\right] \right\|_2^2 \quad (2)$$

Here, $sg(\cdot)$ denotes the stop gradient function, the first term represents the *action reconstruction loss* regulated by a weight factor $w(s_t, a_t)$, and the second term is the *commitment loss* used to regularize the encoder to output vectors close to the embedding vector in in the codetable, $\lambda$ is a factor determines the relative weight, set as 0.25 in our experiments. We do not include an embedding loss, given our employment of a fixed diversity-aware codetable.

**Latent action confidence predictor.** During training, the total number of possible latent actions $K$ in the codetable remains fixed, but the valuable actions can vary across different states. Consequently, we need to set $K$ sufficiently large to encompass all potential actions, which results in redundant latent actions for some states. Furthermore, in the online setting, the action model is dynamically trained. This could potentially lead to substantial fluctuations in the corresponding raw action for the same latent action $k$, resulting in inaccurate estimations of the corresponding Q-values or policy logits outputting by RL agents, often leading to overestimation and subsequent instability issues for both offline and online RL. As depicted in Figure 3, in the offline RL, there may exist some redundant actions for different states that don't belong to offline datasets in LunarLander envrionment. While for online RL, the raw action corresponding to the second latent action changes from A to Z at iteration $t_2$. The real Q-value for Z is 1.2, which is not optimal. However, due to the lag in updating the estimated Q-value, the second latent action, which maximizes the estimated Q-value, is selected for training if we just use the normal DQN. This shift of actions that maximizes Q-value can lead to extra over-optimization problems.

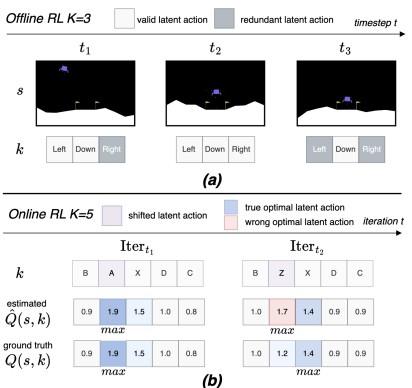

**Figure 3** The *pathological latent action space* with redundant and shifted latent actions. The raw actions corresponding to $k$ are encapsulated in the squares. **Upper**: the latent action exhibits redundancy under different states in an episode. **Lower**: the latent action may shift during training iterations, resulting in wrong Q-value and *argmax* results.

To address these challenges, we propose the learning of a *Latent Action Confidence Predictor*, denoted as $C_\zeta$. This predictor is designed to discern whether each latent action is out-of-distribution (OOD). It takes a state as input and returns a $K$-dimensional confidence score vector for each latent action specific to that state, i.e., $\mathbf{C} = C_\zeta(s)$, where $\zeta$ represents its trainable parameters. Training occurs on the transformed dataset $\{s_t, k_t\}$ using the action encoder of *GADM*. However, direct training of this predictor using simple cross-entropy loss confronts a *data imbalance issue* due to the uneven distribution of data across different states and latent actions. To address this issue, we employ Focal Loss Lin et al. (2017), designed to enhance learning from minority class samples, and initialize the weight bias of the predictor's final layer to be nearly uniform at the start of training. As training progresses, the confidence for the $(s_t, k_t)$ pairs in the dataset steadily increases, while the confidence for out-of-distribution latent actions diminishes. This allows us to set $\frac{1}{K}$ as the *confidence threshold* $\beta$ for identifying OOD actions – actions with confidence less than this threshold are considered OOD. Note that in subsequent discussions with clear context, we often omit the term "latent" from "OOD latent actions". The Loss for training $C_\zeta$ is defined as follows:

$$L_{\text{focal}} = -(1 - sg[p_\text{t}])^\gamma \log(p_\text{t}) \quad (3)$$

where, $p_\text{t}$ represents the confidence assigned by the predictor $C_\zeta$ to the true class (the latent action corresponding to the raw action), i.e., $p_t = C_\zeta(s)[k_t]$, $\gamma$ serves as the focusing parameter, as described in Lin et al. (2017).

**Latent action mask.** After training $C_\zeta$, we can compute the confidence score for each latent action $\mathbf{C}$. Given the *confidence scores* $\mathbf{C}$ and a *confidence threshold* $\beta$, we introduce a *latent action mask* $\mathbb{M}$.

$$\mathbb{M} = I(\mathbf{C} > \beta) \tag{4}$$

Here, $I$ is the indicator function that returns 1 when the condition is true, and 0 otherwise. This means that the mask assigns a value of 1 to the latent actions with confidence scores exceeding $\beta$, and 0 to the remaining actions. This mask can then be seamlessly integrated into standard RL algorithms, enabling the exclusion of unstable, redundant latent actions during training.

### 3.3 PRACTICAL ALGORITHMS

In this section, we demonstrate the versatility of the GADM framework by incorporating it into a model-free method DQN Mnih et al. (2015), and MuZero Schrittwieser et al. (2019), a method rooted in model-based planning. As a result, we derive two instances of our framework, namely, *GADM+DQN* and *GADM+MuZero*, showcasing the adaptability of our framework in varied contexts.

We incorporate the aforementioned *latent action mask* technique into the DQN and MuZero training pipelines to alleviate instability issues. This is the only change required for standard RL algorithms. And it can be easily implemented in the original algorithm pipeline. For example, during the computation of the target Q in DQN, we only consider the latent actions that are not masked by the latent action mask, as these are the latent actions present in the dataset/replay buffer, thus ensuring their Q-values are relatively accurate and meaningful. Specifically, in *GADM+DQN*, the latent action mask TD loss $L_{mask}$ is formulated as follows:

$$L_{mask} = \left[ Q(s_t, k_t; \theta) - [r_t + \gamma Q(s_{t+1}, k_{t+1}; \hat{\theta})] \right]^2 \tag{5}$$

$$k_{t+1} = \underset{k \in M(s_{t+1})}{\arg\max} \ Q(s_{t+1}, k; \theta) \tag{6}$$

where $\theta$ is the parameters of current Q netowrk and $\hat{\theta}$ is the parameters of target Q netowrk, $k_{t+1}$ denotes the latent action at the next time step, chosen from the set of unmasked latent actions $M(s_{t+1})$, which is derived from the latent action mask $\mathbb{M}$,

In the case of MuZero, the latent action mask is directly applied as the *legal action set* at each decision node, which means the MCTS process only considers valid latent actions, enhancing stability.

It is worth highlighting that we do **not** resort to additional techniques commonly used in offline RL, such as conservative value estimation (Kumar et al., 2020) or policy constraints (Fujimoto et al., 2019). Significantly, we are able to address prevalent *pathological latent action space* issues exclusively through the application of the latent action mask technique just in standard RL algorithms like DQN.

## 4 EXPERIMENTS

In this section, we will conduct a thorough evaluation and analysis of the efficiency and scalability of our proposed framework. Our experimental evaluation focuses on the following questions:

- How does *GADM+DQN* compare to baseline algorithms across various action spaces in both online and offline RL benchmark environments? (Section 4.1.1 and Section 4.1.2)

- How does the performance of *GADM* demonstrate when integrated with other decision-making techniques for discrete actions, such as the powerful model-based planning algorithm MuZero Schrittwieser et al. (2019)? (Section 4.1.3)

- What are the influences of various algorithm techniques (e.g. latent action confidence predictor) we proposed in Section 3.2 and Section 3.1? And how do key hyper-parameters and design choices influence the results? (Section 4.1.3)

- What characteristics does the learned latent action space exhibit, and does it encapsulate meaningful high-level semantic information? (Appendix A.6.2 and A.6.1)

For detailed information on benchmark environments, neural network model structures, and hyper-parameter settings, please refer to Appendices A.3, 8, and A.4.3, respectively.

| Dataset | Environment | BC | BCQ | CQL | ICQ | DT | TT | GADM+DQN |
|---------|-------------|-----|-----|-----|-----|-----|-----|----------|
| Medium-Expert | HalfCheetah | 55.2 | 64.7 | 91.6 | **110.3** | 86.8 | 95.0 | 96.4 |
| Medium-Expert | Hopper | 52.5 | 110.9 | 105.4 | 109.0 | 107.6 | 110.0 | **111.4** |
| Medium-Expert | Walker2d | 107.5 | 57.5 | 108.8 | 98.9 | 108.1 | 101.9 | **109.5** |
| Medium | HalfCheetah | 43.1 | 40.7 | 44.0 | 42.5 | 42.5 | 46.9 | **63.3** |
| Medium | Hopper | 52.9 | 54.5 | 58.5 | 55.6 | **67.6** | 61.1 | 58.8 |
| Medium | Walker2d | 75.3 | 53.1 | 72.5 | 71.8 | 74.0 | **79.0** | 74.4 |

**Table 1 (Offline RL)** Benchmark of *GADM+DQN* on the **continuous** locomotion suite of D4RL v2. Competitive performance with various outstanding offline RL algorithms highlight the potential of *GADM*. The evaluation metric is the average normalized score over 4 seeds (Fu et al., 2020b).

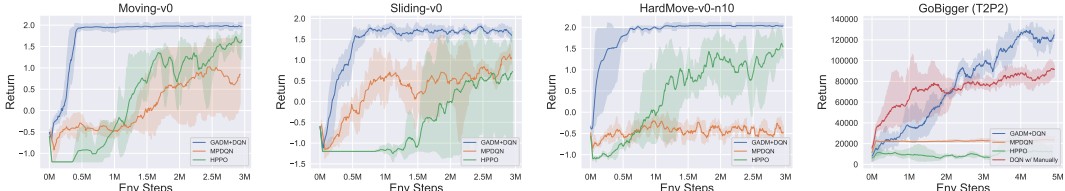

**Figure 4 (Online RL)** Benchmark in four **hybrid** action environments: *Moving*, *Sliding*, *HardMove*, and *GoBigger*. Our method *GADM+DQN* demonstrates both higher performance and greater stability over baselines. Curves and shadings denote the mean and standard deviation return over 10 seeds.

## 4.1 MAIN RESULTS

### 4.1.1 ONLINE RL RESULTS

**Setup** Our main purpose involves testing the performance and stability of our framework across various environments in **Online RL** setting, and comparing it to algorithms specifically tailored for such action spaces. We selected several environments with hybrid or continuous action spaces to validate the effectiveness of *GADM+DQN*. This subsection primarily showcases the experimental setup and results in environments with hybrid action space. We select four representative benchmark environments with **hybrid** action space: *Moving* thomashirtz (2021), *Sliding*, *HardMove* Li et al. (2021), and *GoBigger* Zhang (2021). These environments necessitate the handling of complex relationships within actions. For instance, in *Moving* and *Sliding*, the value of action parameters $y$ is dependent upon the selection of action types $x$. Additionally, it is important to mention that we have chosen *HardMove-v0-n10* as our test setting. This environment's original action space includes both discrete actions and continuous parameters (x, y). Specifically, x encompasses $2^{10}$ discrete actions, and y denotes a ten-dimensional continuous action. This extensive action space presents a significant challenge to conventional hybrid algorithms. For the baseline algorithm, we select MPDQN Bester et al. (2019) and HPPO Fan et al. (2019), which are specifically designed for hybrid action spaces and have demonstrated superior performance within this category. Noted that the line labeled "DQN w/ manually" in *GoBigger* result refers to a sophisticated discretization method in Zhang (2021). For the experimentals within the continuous action space of MuJoCo, please refer to the Appendix A.5.1.

**Results** As illustrated in Figure 4, our experiments substantiate that *GADM+DQN* consistently outperforms baselines MPDQN and HPPO in terms of performance and stability. This underscores the promising potential of representation learning methods such as *GADM* in addressing the intricate relationships within hybrid action spaces. As detailed in Appendix A.2, we attribute the marked performance of *GADM+DQN* to its fundamental approach of discretizing the action space. This strategy simplifies the original action space by eliminating redundant elements. For example, in the *Moving* environment, the decision-making process remains unaffected by any value of action parameters $y$ once the action type $x$ is set to 'break'. This process of streamlining and decoupling significantly enhances the efficiency and stability of the reinforcement learning stage.

### 4.1.2 OFFLINE RL RESULTS

**Setup** To thoroughly assess the performance of *GADM* under **Offline RL** settings, we select the widely used D4RL dataset for investigation. Following previous studies, we focused our experiments on the locomotion subset, comprising the *Walker2D*, *Hopper*, and *HalfCheetah* environments with two different dataset settings: *Medium-Expert and Medium*. We compared our algorithm with several existing methods, including both model-free and model-based offline RL techniques.

| Dataset | Latent Action Shape | GADM+DQN | | | |
|---|---|---|---|---|---|
| | $K$ | $\beta$=1/K | $\beta$=1.5/K | $\beta$=0.5/K | w/o *latent action mask* |
| Hopper Medium-Expert | 4 | **111.4** | 76.2 | 87.3 | 58.0 |
| | 8 | **105.2** | 76.6 | 82.7 | 56.3 |
| Hopper Medium | 4 | **58.8** | 54.6 | 57.6 | 32.4 |
| | 8 | **57.6** | 56.6 | 53.4 | 33.5 |

**Table 3 (Offline RL)** Ablation results for *GADM+DQN* in D4RL v2. The results reveals the significant influence of *latent action confidence predictor* and its mask threshold $\beta$ on the final performance. Neglecting to use the *latent action mask* leads to a drastic deterioration in performance, while even slight adjustments to the threshold can yield moderate impact.

**Results** Experiments in Table 1 reveal that *GADM+DQN* method not only surpasses several classic offline RL algorithms but also performs comparably to Trajectory Transformer (TT) method Janner et al. (2021). It's worth highlighting that in offline settings, *GADM+DQN* is able to tackle *pathological latent action space* issues solely through the *latent action mask* technique and we **did not** employ any additional techniques commonly used in offline RL, such as conservative value estimation Kumar et al. (2020) or policy constraints Fujimoto et al. (2019). This outcome strongly implies the potential of the action discretization model *GADM* in addressing offline OOD issues.

### 4.1.3 GADM + MuZero

| Dataset | GADM+ MuZero | Sampled MuZero |
|---|---|---|
| Medium-Expert | **113.4** | 12.3 |
| Medium | **57.7** | 11.4 |

**Table 2** Offline RL Performance comparison of *GADM+MuZero* and Sampled MuZero in two offline datasets of *Hopper* environment of D4RL v2 with continuous action space.

**Setup** From the first principle, *GADM* is versatile and can be integrated with any algorithms designed for discrete actions. In this subsection, we illustrate this by ingeniously incorporating model-based RL algorithm MuZero. This amalgamation extends the applicability of MuZero to continuous action spaces with offline RL settings. For comparative purposes, we utilize Sampled MuZero (Hubert et al., 2021), an extension of MuZero based on sampled policy iteration, and evaluate their performance in D4RL.

**Results** Table 2 demonstrates that, in an offline scenario, the new algorithm instance *GADM+MuZero* performs well in these two datasets, while Sampled MuZero suffers from noticeably poor performance due to sample efficiency and out-of-distribution issues. This underscores the compatibility of *GADM* with various discrete decision techniques and its potential in addressing offline RL problems.

### 4.2 Ablation Studies

**The impact of latent action mask.** Offline RL can often encounter latent action OOD problems as discussed in 3.2, potentially leading to severe performance degradation. To mitigate this issue, in Section 3.2, we propose the *latent action mask* technique. From our preceding analysis, we proposed that a mask threshold of $\frac{1}{K}$ is a reasonable choice. In this section, we will delve further into the effects of the latent action mask on offline RL performance and explore the implications of varying mask threshold values. Consequently, we designed experiments involving three datasets within the Hopper environment. We evaluated the performance of the confidence threshold, $\beta$, set at $\frac{1}{K}$, $\frac{1.5}{K}$ and $\frac{0.5}{K}$ respectively, as well as the variant algorithm without this technique. These evaluations were conducted under the conditions of K=4 and K=8, allowing us to gain a comprehensive understanding of the impact of different mask threshold values on offline RL performance. Table 3 demonstrates that without the *latent action (confidence) mask* enabled, the performance significantly deteriorates due to the overestimated Q-values for the redundant latent actions. Additionally, even minor adjustments to the confidence threshold—either increases or decreases—can have a substantial influence on model performance. This underscores the criticality of selecting 1/K as the optimal confidence threshold.

**The impact of latent action remapping and warmup.** We further examine the influence of improvement techniques proposed in Section 3.1 for *GADM* on *HalfCheetah-v3* and *HardMove-v0-n10*. Specifically, we have two ablation variants: *(1): **GADM+DQN w/o Latent Action Remapping**: This variant does not involve the remapping of latent actions during the RL training phase. (2): **GADM+DQN w/o Warmup**: This variant initiates RL training without the warmup stage.* Figure 5 show that when we remove either of proposed techniques, the performance of *GADM+DQN* drops significantly in both two environments, verifying the effectiveness of our proposed techniques.

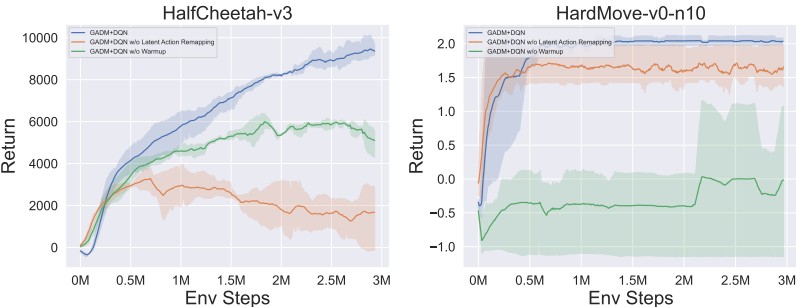

**Figure 5** Ablation results for *GADM+DQN* in environments with continuous and hybrid action spaces. There is significant drop when any of our proposed improvement techniques are removed.

Beside, the influence of varying latent action shapes $K$ and state-conditions is elucidated in Appendix A.5. To further analyze the learned space, we also conduct several visualizations in Appendix A.6.1.

## 5 RELATED WORK

RL encompasses numerous problems characterized by complex action spaces. For instance, some problems exhibit high-dimensional continuous action space. Within such spaces, the presence of countless actions poses challenges for exploration (Dalal et al., 2021) and neural network optimization (Bjorck et al., 2021). Conversely, some problems feature hierarchical action space structures (Wei et al., 2022), where the selection of actions becomes increasingly intricate. Moreover, actions in hybrid spaces comprise a combination of discrete and continuous actions, necessitating the specialized network design. In summary, training RL agents for complex actions presents a significant challenge.

To address the issues of action spaces, researchers have proposed various customized algorithms that allow RL agents to learn directly in these raw action spaces. For instance, Parameterized Action DDPG(Hausknecht & Stone, 2016) employs a modified DDPG actor-critic structure to handle multiple action parts, while HPPO (Fan et al., 2019) improve original PPO to deal with hierarchical action structures. To further combine the advantages of distinct common RL algortihms, PDQN (Xiong et al., 2018) and MPDQN (Bester et al., 2019) utilize a hybrid structure of DQN and DDPG, explicitly modeling dependencies between continuous and discrete sub-actions.

Action discretization is another possible solution. However, it needs the domain knowledge about the specific environment and often encounters the curse of dimensionality in actions (Kanervisto et al., 2020). To mitigate these problems, Tang & Agrawal (2020) verifies the feasibility of discretizing the action space in on-policy optimization. The study conducted by (Seyde et al., 2021) explores the impact of extreme actions on continuous control. These works also inspire the design of *GADM*, including network architectures and optimization tricks.

Action representation learning offers another promising idea to adaptively discretize action spaces. Chandak et al. (2019) proposes representation learning in a large action space, leveraging the structural characteristics of actions and demonstrating its significance in enhancing generalization in real-world applications. Pritz et al. (2020) attempts to utilize the joint-learned state-action embeddings to boost decision-making. Li et al. (2021) proposes to learn a compact and decodable latent representation space for the original hybrid action space. HyAR constructs the latent space and incorporates the dependence between action parts through an embedding table and conditional VAE. Dadashi et al. (2022) and Gu et al. (2022) propose to learn a set of plausible discrete actions from expert demonstrations to overcome the curse of dimensionality problem. Conversely, we introduce a unified decision-making framework. It is designed to learn compact, discrete latent actions in environments with diverse action spaces and stands as the first decoupling paradigm for both online and offline RL.

## 6 CONCLUSION AND LIMITATION

Starting from the comprehensive analysis for action spaces designs, we propose a novel action representation learning algorithm *GADM* adapted to common RL algorithms, which can be a unified and efficient paradigm for diverse decision-making scenarios. Although our method achieve superior performance in different benchmark environments, there are still some challenging problems, such as variable-length actions in episodes and large-scale discrete actions in language models. Besides, learning latent actions from the viewpoint of conditional sequential generation is also a valuable attempt. We will continue to pursue the ultimate solution for action space in future work.

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

# A APPENDIX

## A.1 DETAILS OF BACKGROUND

In this section, we first offer an general definition of the Hybrid Action Space. Subsequently, we present a formal explanation of the Action Transformed MDP that appears in Figures 1 and 2 in the paper.

**Hybrid Action Space** This work (Masson et al., 2016) introduces the parameterized action space, which is an hierarchical action space with two layers. Specifically, $\mathcal{A}_d$ represents a finite set of discrete actions, while $\mathcal{X}_a$ denotes the continuous parameter space associated with action $a \in \mathcal{A}_d$. During action selection, we first choose a discrete action $a$ from $\mathcal{A}_d$, and then select a parameter $x$ from the corresponding space $\mathcal{X}_a$. A complete action is uniquely determined by a binary tuple $(a, x)$. Namely, the action space is then given by:

$$\mathcal{A} = \{(a, x) \mid a \in \mathcal{A}_d, x \in \mathcal{X}_a\} \tag{7}$$

To represent a more general hybrid action space, as found in complex video games like StarCraftII (Vinyals et al., 2017) or Dota2 (OpenAI, 2018), we extend the two-layer structure to $N$-layer. We decompose an action into $N$ successive selections across different layers. In the initial layer of selection, we choose a node $a_1$ from the initial set $\mathcal{A}_0$. As we progress to the $i$-th layer of selection, the previous selection sequence $a_{1:i-1} = (a_1, a_2, ..., a_{i-1})$ defines a corresponding space $\mathcal{X}_{a_{1:i-1}}$. Subsequently, we select a node $a_i$ from $\mathcal{X}_{a_{1:i-1}}$. A complete action is uniquely determined by an $N$-tuple $(a_1, a_2, ..., a_N)$. Furthermore, the action space can be described as:

$$\mathcal{A} = \{(a_1, a_2, ..., a_N) \mid a_1 \in \mathcal{A}_0, a_i \in \mathcal{X}_{a_{1:i-1}} \text{ for } i = 2, ..., N\} \tag{8}$$

Notably, parameterized action space is a special case of our definition where $N = 2$ and $\mathcal{A}_0 = \mathcal{A}_d$. A common hybrid action space in everyday life is the realm of keyboard shortcuts. Let's take the shortcut Ctrl + Shift + Esc as an example. To utilize this shortcut, we begin by pressing the Ctrl key on the keyboard. This initiates a candidate set comprising all the keys that can be combined after Ctrl. From this set, we choose Shift as the second key. Once again, the various options that can be paired with Ctrl + Shift form a set of choices. Finally, we select Esc from the choices as the third key. This process exemplifies the execution of actions within a hybrid action space with $N = 3$.

**Action Transformed MDP** We introduce a transformation operator $\mathcal{T}$, which can transform the raw action into the latent space. Denote the transformed action as $k$, we can describe $\mathcal{T}$ as:

$$\mathcal{T} : k = \mathcal{T}(s, a) \tag{9}$$

And we can reconstruct an action in the raw action space from the latent space through $\mathcal{T}^{-1}$:

$$\mathcal{T}^{-1} : a' = \mathcal{T}^{-1}(s, k) \tag{10}$$

In fact, $\mathcal{T}$ may map different actions from the raw action space to the same $k$ in the latent space. These actions often have similar effects in the environment, such as resulting in the same state transition and obtaining the same reward. Therefore, $\mathcal{T}^{-1}$ is not a strict inverse transformation of $\mathcal{T}$. That is to say, $a'$ may not necessarily be the same as the original $a$, but it will have a similar effect in the environment. With the transformation $\mathcal{T}$ and $\mathcal{T}^{-1}$, we can convert an original MDP $\mathcal{M} = (\mathcal{S}, \mathcal{A}, \mathcal{P}, \mathcal{R}, \gamma, \rho_0)$ into an action transformed MDP $\mathcal{M}_{\mathcal{T}} = (\mathcal{S}, \mathcal{K}, \mathcal{P}_{\mathcal{T}}, \mathcal{R}_{\mathcal{T}}, \gamma, \rho_0)$, where $\mathcal{K}$ is the latent action space, which can be defined by:

$$\mathcal{K} = \{k \mid k = \mathcal{T}(s, a), \text{for all } (s, a) \text{ in } \mathcal{S} \times \mathcal{A}\} \tag{11}$$

And $\mathcal{P}_{\mathcal{T}}$ and $\mathcal{R}_{\mathcal{T}}$ are the transition function and reward function, respectively, induced by $\mathcal{P}$ and $\mathcal{R}$. Specifically, they are defined as follows:

$$\mathcal{P}_{\mathcal{T}}(\cdot \mid s, k) = \mathcal{P}(\cdot \mid s, \mathcal{T}^{-1}(k)) \tag{12}$$

$$\mathcal{R}_{\mathcal{T}}(\cdot \mid s, k) = \mathcal{R}(\cdot \mid s, \mathcal{T}^{-1}(k)) \tag{13}$$

The other elements are consistent with the original definition of $\mathcal{M}$.

### A.2 MOTIVATION

In this section, we provide a thorough discussion of the foundational motivations and insights that inspired the design of *GADM*.

#### A.2.1 UNIFIED AND DECOUPLING ACTION REPRESENTATION LEARNING

The diversity of raw action spaces is one of the most challenging aspect for RL training. To improve data efficiency and training stability, it is inevitable to first conduct different action shaping operations and select specific algorithms for the corresponding action spaces, which brings non-negligible learning and tuning costs beyond core RL optimization. Therefore, some researchers begin to find the solution to learn general and compact action representations adaptively instead of repeating some dirty work in the raw action space. Moreover, since RL agents can be directly applied into the learned discrete action space, this scheme can be seen as a kind of decoupling of action representation learning and RL, which allows researchers to concentrate on only one of the topics. For training pipeline, these two parts can be alternative or parallel training according to concrete situations.

#### A.2.2 ACTION SPACE REDUNDANCY

Action space redundancy in reinforcement learning mainly refers to the presence of multiple actions that lead to the same state transition or reward. This means that there may be more than one way to achieve the desired result using different actions. The existence of redundant actions may cause agents to explore many invalid or useless actions instead of necessary valuable actions, making it challenging for agents to learn optimal policies efficiently.

This redundancy widely exists in discrete, continuous, and hybrid action spaces. In discrete action space, redundancy is reflected in the fact that the state transitions and rewards caused by different actions are similar or even completely consistent. As shown in Figure 1 (left), there is a lot of redundancy in the raw actions in the Sokoban (Schrader, 2018) environment. For continuous control, action redundancy manifests itself in the fact that small changes between actions do not have a large impact on future returns. For example, the difference of Q value between an action and its nearby actions during training of TD3 (Fujimoto et al., 2018) algorithm will not be very large (Figure 1 (right)). As shown in equation (2), the redundancy of the hybrid action space lies in the existence of a large number of illegal actions, that is, $a_i \notin \mathcal{X}_{a_{1:i-1}}$. This requires designing complex mechanisms for specific environments to ensure efficient exploration and steady policy optimization.

#### A.2.3 THE INFLUENCE OF OFFLINE DATASET

To better distinguish between the necessary and redundant subsets of the original action space, especially in some complex action spaces, it is reasonable to combine action represnetation learning with the offline dataset. Since the purpose of RL agents is not to traverse all states and actions of the environment but to find the optimal decision-making policies, proper offline datasets can provide valuable initial experiences for action representation learning, making it easier to take high-value actions instead of random exploration in the following training. Furthermore, state-action pairs in datasets can not only serve as proper action candidates, but also be considered as an implicit constrain, which are similar to the idea proposed in recent offline RL algorithms BCQ Fujimoto et al. (2019) and ICQ Yang et al. (2021). Due to the lack of online interaction with environments, offline RL algorithms usually need to be conservative enough to avoid the out-of-distribution problem of target actions in TD learning. The action model pretrained on existing offline datasets can be an appropriate constraint design for all kinds of action spaces (Section 4.1.2).

### A.3 BENCHMARK ENVIRONMENTS

In this section, we provide detailed descriptions of benchmark environments used in our experiments.

**D4RL** Fu et al. (2020a) is a widely used offline RL benchmark that comprises several environments and datasets. In line with numerous prior studies, we concentrate on the locomotion subset: *Hopper*, *HalfCheetah*, and *Walker2D*. For each environment, we evaluate three different dataset configurations: *Medium-Expert*, *Medium*, and *Medium-Replay*, which facilitate the study of offline RL.

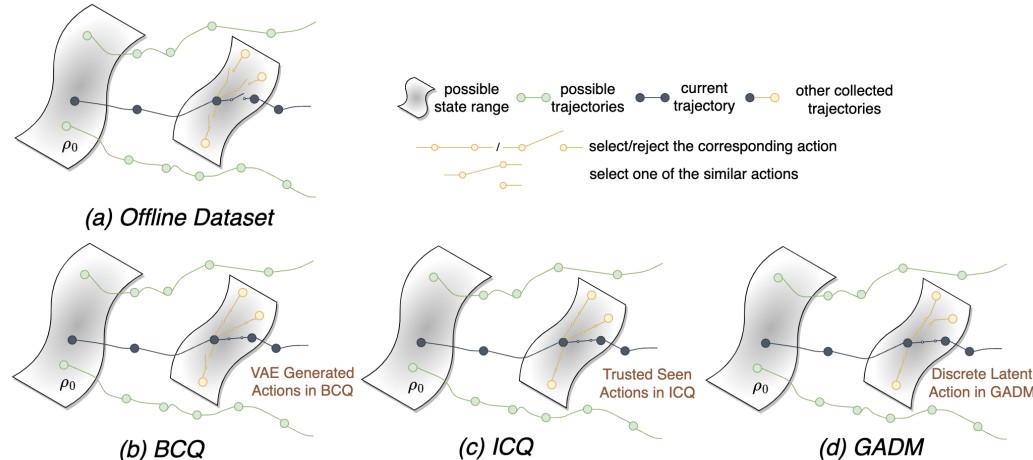

**Figure 6** Comparison of different action constraints across various offline RL algorithms. BCQ Fujimoto et al. (2019) is only capable of generating certain parts of the original action space, while ICQ Yang et al. (2021) performs optimization on all trusted seen actions. In contrast, *GADM* can learn more compact spaces than the others.

**MuJoCo** stands for **mu**lti-**jo**int dynamics with **co**ntact. It is a versatile physics engine designed to support research and development in fields such as robotics, biomechanics, graphics and animation, machine learning, and others that necessitate rapid and precise modeling of articulated structures interacting with their environment Todorov et al. (2012). The *MuJoCo* environment features continuous action spaces and reward representations composed of multiple components, often including penalties for actions corresponding to poor control. We evaluate our proposed *GADM+DQN* along with other baseline algorithms in: *Hopper-v3*, *HalfCheetah-v3,*, *Ant-v3*, and *Humanoid-v3*.

**Gym-Hybrid** [1] constitutes a range of sandbox environments with parameterized (also called hybrid) action space. The objective of the agent is to halt within a target area, a circle with a radius of 0.1 situated within a square field with a side length of 2. There are **three** discrete actions: *turn, accelerate, and brake*, complemented by **two** possible parameters: *acceleration and rotation*. We also employ the *HardMove-v0-n\** introduced in (Li et al., 2021), where the agent controls $n$ evenly spaced actuators, choosing whether each should be on or off and determining the corresponding continuous parameter for each actuator (moving distance) to reach the target area. An increase in $n$ corresponds to a larger action space and a greater challenge for the agent in terms of exploration and learning. In our experiments, we use the hardest case that *n=10*.

**GoBigger** is a multi-agent RL environment emphasizing cooperation and competition. Each agent, represented by one or multiple balls (termed clone balls), enlarges its size by colliding and merging with other balls within a bounded rectangular area within a fixed time frame. A larger clone ball size corresponds to a higher score. The observation space in *GoBigge*r includes information about all units within the agent's local field of view, and the reward is calculated as the difference in sizes across two consecutive timesteps. The action space is a hybrid one *(x, y, action_type)*, similar to *Gym-Hybrid*. Given the rapid development or elimination of opponents through continuous cooperation in *GoBigger*, precise actions are usually required for cooperation, making action representation a significant challenge. *GoBigger* includes multiple sub-environments that can be tailored for various tasks. Commonly used environments include *t2p2*, *t3p2*, *t4p3*, with the number following *t* (team) indicating the number of teams in a game, and the number following *p* (player) indicating the number of agents per team. In our experiments, we configure the parameters such that *t=p=2*. Given that this is a multi-agent environment, we employ an independent learning mechanism (de Witt et al., 2020) to facilitate the adaptation of *GADM+DQN* to this context. Correspondingly, similar mechanisms are utilized for comparative algorithms to ensure a fair evaluation.

---

[1]https://github.com/thomashirtz/gym-hybrid

In Figure 7, we have visualized two benchmark environments that feature complex hybrid action spaces, HardMove-v0-n* and GoBigger (T2P2). These visualizations provide a graphical representation of the challenges and complexities inherent in these environments, thereby highlighting the capabilities required of effective action representation and learning models.

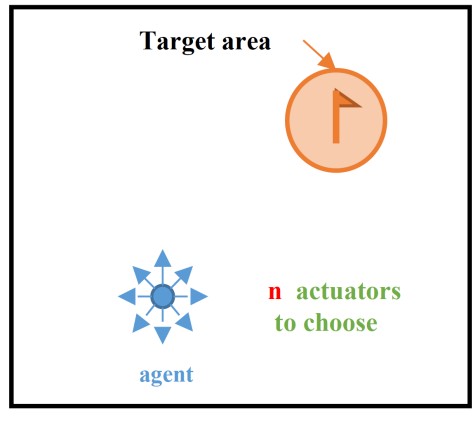

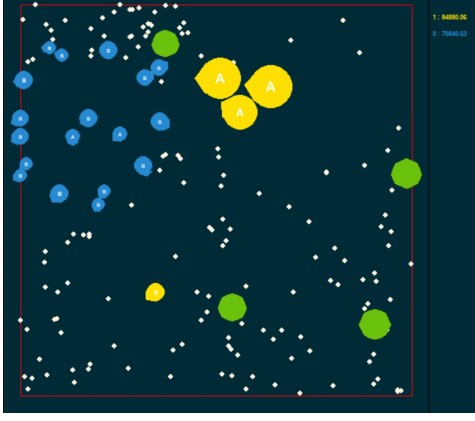

(a) HardMove-v0-n*                    (b) GoBigger (T2P2)

**Figure 7** Benchmark Environments featuring complex hybrid action spaces: (a) Illustrates an *HardMove-v0-n\** environment where the agent's goal is to navigate to the target area. The agent is equipped with $n$ uniformly distributed actuators, with the capability to decide the status (on or off) of each actuator while simultaneously determining the corresponding continuous parameters. (b) Depicts a dynamic multi-agent arena in *GoBigger (T2P2)*, each agent, represented by a ball, aims to increase its size (or weight) by strategically colliding and merging with other balls within a bounded rectangular space, all within a specified time limit (e.g. 300 seconds). The same colors mean the same team while the distinct alphabets mean different players.

## A.4 IMPLEMENTATION DETAILS

In this section, we begin by outlining the pseudocode for our proposed framework, which is applicable to both online and offline reinforcement learning environments. We then dig deeper into the architecture of the GADM model, with a particular focus on elucidating the functionalities of the action model. Furthermore, we describe the hyperparameters associated with *GADM+DQN*, *GADM+MuZero*, and the baseline algorithms across a range of different settings. Concluding this section, we detail the computational overhead that our algorithm incurs, thereby providing a holistic view of its performance characteristics.

### A.4.1 PSEUDOCODE

Algorithms 1 and 2 present the seamless integration of the *GADM* model with traditional reinforcement learning algorithms in both online and offline contexts. We plan to release our source code publicly following the conclusion of the full review process.

Algorithm 1 explores the confluence of the *GADM* algorithm with online reinforcement learning situations. The algorithm splits fundamentally into two main *Stages*: an optional *Warmup Stage* and an *Online Stage*. These stages encompass various *Phases*, including data collection, action representation learning (AR), reinforcement learning (RL), and latent action confidence prediction (LACP). Algorithm 2 lays out the methodological structure of the *GADM* algorithm when applied in offline reinforcement learning scenarios. This algorithm is built around three central *Phases*: action representation learning (AR), latent action confidence prediction (LACP), and the actual reinforcement learning phase (RL).

Furthermore, inspired by our practical experiments and insights from previous research Seyde et al. (2021), we discovered that for environments with continuous action spaces, such as MuJoCo, the endpoint values of the action threshold bear a significant impact on performance. To tackle the challenges associated with modeling extreme action values, we devised a technique known as

Extreme Action Regression (E.A.R.) to handle the need for extreme values in certain environments. Notably, this technique is applicable not only to environments requiring specific extreme action values but also to generic environments.

---

**Algorithm 1** GADM for Online RL

---

1: Initialize *GADM*: Encoder $e_\phi$, Decoder $d_\psi$, Action embedding table $V_\varepsilon$
2: Initialize action representation (AR) buffer $\mathcal{D}_{\text{AR}}$, reinforcement learning (RL) buffer $\mathcal{D}_{\text{RL}}$
3: Initialize RL agent networks (such as $Q_\theta$ and/or $\pi_\omega$)
4: Initialize exploartion strategies $\chi_{\text{RL}}, \chi_{\text{AR}}$
5: ① *Warmup Stage (optional)*
6: **while** not reaching maximum warm-up environment steps **do**
7:     // Warm-up data collection phase
8:     **while** not reaching maximum warm-up collecting steps **do**
9:         Collect data using random/expert policy and store $\{s_t, a_t, r_t, s_{t+1}\}$ in buffer $\mathcal{D}_{\text{AR}}$
10:     // Warm-up AR training phase
11:     **while** not reaching maximum warm-up training steps **do**
12:         Sample mini-batch $\{s_t, a_t\}$ from $\mathcal{D}_{\text{AR}}$
13:         Update $\phi, \psi$ and $\varepsilon$ using the sampled mini-batch
14: ② *Online Stage*
15: **while** not reaching maximum online environment steps **do**
16:     // Online data collection phase
17:     **for** t $\leftarrow$ 1 to $T$ **do**
18:         $k_t = \chi_{\text{RL}}(\text{agent}(s_t))$ // Select latent action in the latent action space
19:         $a_t = \chi_{\text{AR}}(d_\psi(s_t, k_t))$ // Decode into raw action space
20:         Execute raw action $a_t$, observe reward $r_t$ and next state $s_{t+1}$
21:         Store $(s_t, a_t, r_t, s_{t+1})$ in $\mathcal{D}_{\text{RL}}$ and $\mathcal{D}_{\text{AR}}$
22:     // Online AR training phase
23:     **while** not reaching maximum AR training steps **do**
24:         Sample mini-batch $\{s_t, a_t, r_t, s_{t+1}\}$ from $\mathcal{D}_{\text{AR}}$
25:         Update $\phi, \psi$ and $\varepsilon$ using the sampled mini-batch
26:     // Online LACP phase (optional)
27:     **while** not reaching maximum LACP training steps **do**
28:         Sample mini-batch $\{s_t, a_t\}$ from $\mathcal{D}_{\text{offline}}$
29:         Update $\zeta$ using Focal Loss $L_{focal}$ on the sampled mini-batch
30:     // Online RL training phase
31:     **while** not reaching maximum RL training steps **do**
32:         Sample mini-batch $\{s_t, a_t, r_t, s_{t+1}\}$ from $\mathcal{D}_{\text{RL}}$
33:         Compute confidence scores $\mathbf{C}$ for all latent actions using $C_\zeta(s_t)$
34:         Determine latent action mask $\mathbb{M}$ based on confidence scores $\mathbf{C}$ and threshold $\beta$
35:         $k_t = e_\phi(s_t, a_t)$ // Latent Action Remapping
36:         Update agent networks using $\{s_t, sg[k_t], r_t, s_{t+1}\}$ and the latent action mask $\mathbb{M}$

---

**Extreme Action Regression** Another innate property of raw action spaces deserving careful consideration relates to special action values, such as the extreme actions pointed out in Seyde et al. (2021), or the action thresholds of engine dynamics in LunarLander (please refer to A.6.2). Precise recreation of these unique continuous values by the action model is essential. For this purpose, in the action decoder of *GADM*, we employ the distributional head (Bellemare et al., 2017) to automatically reconstruct these specific actions.

Assuming the range of the raw continuous action is represented by $[A_{\min}, A_{\max}]$, and it is divided into $N + 1$ equally spaced bins, the action decoder of our model outputs an $N$-dimensional probability distribution. The predicted action can then be obtained as:

$$\hat{a} = \sum_{j=0}^{N} s_j * p_j(s_i), s_j = A_{\min} + j * (A_{\max} - A_{\min})/N \tag{14}$$

---

**Algorithm 2** GADM for Offline RL

---
1: **Given** the offline dataset $\mathcal{D}_{\text{offline}}$, confidence score threshold $\beta$
2: **Initialize** *GADM*: Encoder $e_\phi$, Decoder $d_\psi$, Action embedding table $V_\varepsilon$, Latent action confidence predictor $C_\zeta$
3: **Initialize** RL agent networks (such as $Q_\theta$ for DQN, $Q_\theta$ and $\pi_\omega$ networks for MuZero)
4: Phase ① : Learn the action representation model (AR)
5: **while** not reaching maximum AR training steps **do**
6:     Sample mini-batch $\{s_t, a_t\}$ from $\mathcal{D}_{\text{offline}}$
7:     Update $\phi, \psi$ and $\varepsilon$ using reconstruction loss $L_{rec}$ on the sampled mini-batch
8: Phase ② : Learn the latent action confidence predictor (LACP)
9: **while** not reaching maximum LACP training steps **do**
10:     Sample mini-batch $\{s_t, a_t\}$ from $\mathcal{D}_{\text{offline}}$
11:     Update $\zeta$ using Focal Loss $L_{focal}$ on the sampled mini-batch
12: Phase ③ : Apply any discrete action RL methods (e.g. DQN and MuZero)
13: **while** not reaching maximum RL training steps **do**
14:     Sample mini-batch $\{s_t, a_t, r_t, s_{t+1}\}$ from $\mathcal{D}_{\text{offline}}$
15:     Compute confidence scores **C** for all latent actions using $C_\zeta(s_t)$
16:     Determine latent action mask $\mathbb{M}$ based on confidence scores **C** and threshold $\beta$
17:     $k_t = e_\phi(s_t, a_t)$ // Latent Action Mapping
18:     Update agent networks using $\{s_t, sg[k_t], r_t, s_{t+1}\}$ and the latent action mask $\mathbb{M}$

---

where $p_j(s_i)$ is the $jth$ output probability of the reconstruction distribution of the action decoder for $(s, a)$. During evaluation, if the probability of the support exceeds a specific threshold (for example, 0.9), we directly output the corresponding support value. This design has proven to be particularly effective in certain environments, such as *Hopper and Halfcheetah* in MuJoCo benchmark.

### A.4.2 NETWORK ARCHITECTURE

**Action Model** The network architecture of the action encoder and decoder for *GADM* designed for hybrid action spaces is depicted in Figure 8. As discussed in Section 3.2, the inputs of the action encoder encompass both the state and action embeddings. A skip connection of the state embedding is also integrated into the action decoder. The output of the action decoder bifurcates into two-types of heads, each tasked with reconstructing the continuous and discrete components of the raw hybrid action, respectively. Take the *Gym-Hybrid* environment for instance: the raw action is a tuple composed of (action type, action arguments). In this context, action type is the discrete action that determines the type of the hybrid action, such as {*accelerate*, *turn*, *brake*}. Action arguments are two-dimensional continuous action. Since the dimension of states is usually much larger than those of actions, which may lead to *posterior collapse* (Lucas et al., 2019) issue during training, we leverage the effective feature fusion method FiLM (Perez et al., 2017) to merge state and action embeddings.

**Latent Action Confidence Predictor** The network architecture of latent action confidence predictor $C_\zeta$ is a simple state encoder, which employs a 3-layers MLP (multilayer perceptron) for vector states or a 3-layers convolutional neural network for image states. It takes a state as input and returns a $K$-dimensional confidence score vector for each latent action specific to that state, i.e., $\mathbf{C} = C_\zeta(s)$. Given the *confidence scores* **C** and a *confidence OOD threshold* $\beta$, we can compute a *latent action mask* used for the *latent action mask* technique.

### A.4.3 HYPERPARAMETERS

We provide the hyperparameters for our proposed methods, *GADM+DQN* and *GADM+MuZero*, in an offline setting for D4RL in Table 6 and Table 7, respectively.

In an online setting, we provide the hyperparameters for hybrid action space environments in Table 5, and for continuous action space environments, specifically MuJoCo, in Table 4.

We adhere to the same hyperparameters as provided in the original papers for the continuous action space baseline algorithm TD3, and the hybrid action space algorithms MPDQN Bester et al. (2019), and HPPO Fan et al. (2019). This ensures consistency in our methodological approach.

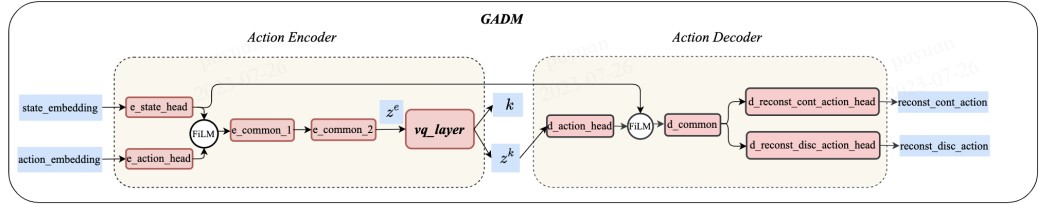

**Figure 8** Network structure of the action encoder and decoder in *GADM* for *hybrid* action spaces. The inputs of the action encoder include both the state and action embeddings. Since the dimension of states is usually much larger than those of actions, which may lead to *posterior collapse* (Lucas et al., 2019) issue during training, we leverage the effective feature fusion method FiLM (Perez et al., 2017) to merge state and action embeddings. A skip connection of the state embedding is also incorporated into the action decoder.

| DQN Hyper-parameter | Value |
|---|---|
| Discount factor | 0.99 |
| Learning rate | 3e-3 |
| DQN replay buffer size | 1e6 (transitions) |
| Hidden size list of Q network | [256, 256, 128] |
| N sample per collect | 256 (transitions) |
| Batch Size | 512 |
| Update per collect | 50 |
| *GADM* Hyper-parameter | Value |
| Learning rate | 3e-3 |
| Hidden size list of encoder | [256, 256,256] |
| Batch Size | 512 |
| The number of embedding vectors (i.e. latent action shape) | 128 (64 for *Hopper-v3*) |
| The dimension of embedding vectors | 256 |
| Warmup data size | 5e4 |
| Warmup update steps | 1e4 |

**Table 4** Key Hyperparameters of *GADM+DQN* used in (continuous action space) *MuJoCo*.

| DQN Hyper-parameter | Value |
|---|---|
| Discount factor | 0.99 |
| Learning rate | 3e-3 |
| RL replay buffer size | 1e6 (transitions) |
| *DQN* replay buffer size | 1e6 (transitions) |
| Hidden size list of Q network for *HardMove-v0-n10* and *GoBigger* | [256, 256, 128] |
| Hidden size list of Q network for *Moving-v0* and *Sliding-v0* | [128, 128, 64] |
| Ensemble Number (N) | 20 |
| N sample per collect | 256 (transitions) |
| Batch Size | 512 |
| Update per collect | 50 |
| *GADM* Hyper-parameter | Value |
| Learning rate | 3e-3 |
| Hidden size list of encoder | [256, 256, 256] |
| Batch Size | 512 |
| latent action shape for *Moving-v0* and *Sliding-v0* | 16 |
| latent action shape for *HardMove-v0-n10* and *GoBigger* | 64 |
| The dimension of embedding vectors | 64 |
| Warmup data size | 5e4 |
| Warmup update steps | 1e4 |

**Table 5** Key Hyper-parameters of *GADM+DQN* on (hybrid action space) *Gym-Hybrid* and *GoBigger*.

| DQN Hyper-parameter | Value |
|---|---|
| Training Epoch | 500 |
| Discount factor | 0.99 |
| Learning rate | 3e-3 |
| Hidden size list of Q network | [256, 256, 128] |
| Batch Size | 512 |
| *GADM* Hyper-parameter | Value |
| Training Epoch | 200 |
| Learning rate | 3e-3 |
| Hidden size list of encoder | [256, 256, 256] |
| Batch Size | 512 |
| Latent action shape | 16 (4 for *Hopper*) |
| The dimension of embedding vectors | 256 |

**Table 6** Key Hyperparameters of *GADM+DQN* used in offline *D4RL* Environment.

| MuZero Hyper-parameter | Value |
|---|---|
| Training Epoch | 500 |
| Optimizer type | Adam |
| Learning rate | $3 \times 10^{-3}$ |
| Discount factor | 0.997 |
| Weight of policy loss | 1 |
| Weight of value loss | 0.25 |
| Weight of reward loss | 1 |
| Weight of policy entropy loss | 0 |
| Weight of SSL (self-supervised learning) loss | 2 |
| Batch size | 512 |
| Frequency of target network update | 100 |
| Weight decay | $10^{-4}$ |
| Max gradient norm | 10 |
| Length of game segment | 200 |
| TD steps | 5 |
| Number of unroll steps | 5 |
| Discrete action encoding type | One Hot |
| Normalization type | Batch Normalization |
| Dirichlet noise alpha | 0.3 |
| Dirichlet noise weight | 0.25 |
| Number of simulations in MCTS (sim) | 100 |
| Reanalyze ratio | 1 |
| Categorical distribution in value and reward modeling | True |
| The scale of supports used in categorical distribution | 300 |
| *GADM* Hyper-parameter | Value |
| Training Epoch | 200 |
| Learning rate | 3e-3 |
| Hidden size list of encoder | [256, 256, 256] |
| Batch Size | 512 |
| Latent action shape | 16 (4 for *Hopper*) |
| The dimension of embedding vectors | 256 |

**Table 7** Key Hyperparameters of *GADM+MuZero* used in offline *D4RL* Environment.

### A.4.4 COMPUTATIONAL COST

All our experiments are performed on the NVIDIA V100 GPU. The experiments on *MuJoCo* environment with continuous action space taken approximately 1.5 hours to achieve training iterations on 3M env steps in each seed. The experiments on *Gym-Hybrid* environments with hybrid action space taken approximately 2 hours to achieve training iterations on 3M env steps in each seed. The

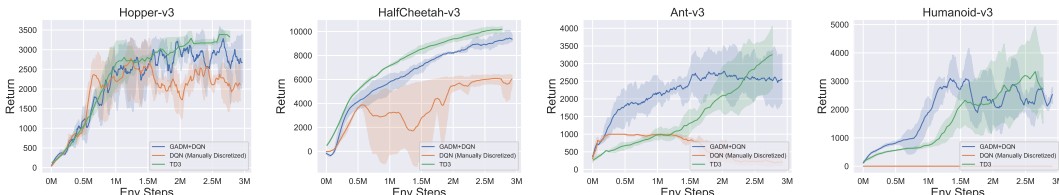

**Figure 9 (Online RL)** Benchmark performance on four **continuous** action environments in *MuJoCo*. Our proposed method, *GADM+DQN*, achieves results on par with TD3 and demonstrates a significant improvement over DQN with naive manually discretized action space across all four domains. Curves and shadings denote the mean and standard deviation over 10 seeds.

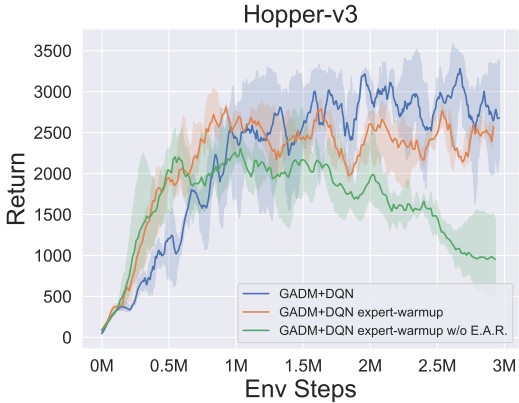

**Figure 10** The Effect of different mechanisms of warmup in our GADM. The x- and y-axis denote the environment steps ($\times 10^6$) and average episode return over 10 episodes, respectively. Curves and shading denote the mean and standard deviation over 3 seeds.

experiments on *GoBigger* environments with hybrid action space taken approximately 8 hours to achieve training iterations on 3M env steps. D4RL experiments usually finish in 30 minutes.

### A.5  ADDITIONAL EXPERIMENTS

#### A.5.1  ONLINE MUJOCO RESULT

In this section, we examine the efficacy and efficiency of GADM across various continuous and hybrid-action environments, contrasting it with previous algorithms specifically tailored for these action spaces. We first test our approach on MuJoCo, a classic benchmark for continuous control. This includes two high-dimensional continuous domains: Ant and Humanoid, with 8 and 17 dimensions respectively. It's worth noting that we introduce several redundant dimensions to the raw action spaces for this experiment. Our *GADM+DQN* is compared against two groups: TD3, a popular algorithm for continuous action spaces, and a basic DQN implemented in a manually discretized action space. In the latter, we evenly divide the raw continuous action into 3 bins at each dimension and use their Cartesian product to derive handcrafted discrete actions. As illustrated in Figure 9, *GADM+DQN* yields results comparable to TD3 and demonstrates a clear enhancement over the basic DQN across all four domains.

#### A.5.2  THE EFFECT OF EXPERT DATA WARMUP

In this section, we investigate the effect of different mechanisms of warmup in GADM+DQN on *Hopper-v3* (continuous). The comparison curves are shown in Figure 10. Concretely, we have the following two variants in total, and their brief descriptions are as follows:

**Collection of Expert Data**   First, we train the TD3 agent until convergence, then use the best TD3 agent interact with the environment to collect 1000 episodes, then we only select the episodes whose return is larger than 3500 as the expert warmup dataset, in total, about 269800 transitions. When we pre-training *GADM*, we set the epochs as 20.

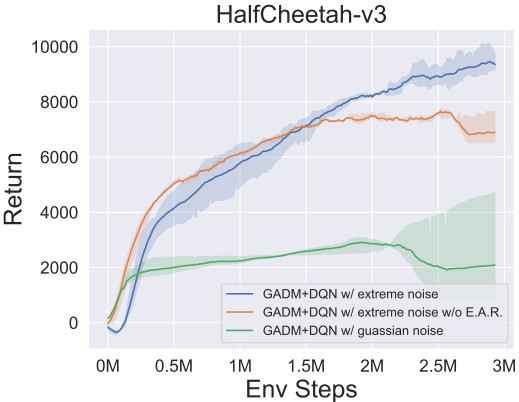

**Figure 11** The Effect of different mechanisms of Exploring the Raw Action Space in our GADM. The x- and y-axis denote the environment steps ($\times 10^6$) and average episode return over 10 episodes, respectively. Curves and shading denote the mean and standard deviation over 3 seeds.

**GADM+DQN-expert-warmup**: The variant agent pretrain the *GADM* utilizing the expert data.

**GADM+DQN-expert-warmup w/ E.A.R.**: The variant agent don't use *Extreme Action Regression* in *GADM*, and *GADM* is pretrained utilizing the expert data.

### A.5.3 DIFFERENT MECHANISMS OF EXPLORING THE RAW ACTION SPACE

There are two places in our *GADM* framework that need to introduce exploration mechanisms, namely exploration in latent action space $\chi_{\text{RL}}$ and exploration in Raw action space $\chi_{\text{AR}}$. Our instance method *GADM+DQN* is essentially value-based, thus naturally, we adapt the usual epsilon-greedy exploration mechanism as $\chi_{\text{RL}}$, i.e. $k_t = \epsilon - Greedy(Q(s_t, .))$. Another core problem is how to efficiently explore the raw action space without affecting the stability of the *GADM* framework.

Motivated by (Seyde et al., 2021), we propose a special noise mechanism, namely, with a small probability (e.g. 0.1), we execute the random Bernoulli extreme action instead of the decoded raw action. And we also experiment the usually gaussian noise mechanism. Therefore, we investigate the effect of different mechanisms of exploring the raw action space $\chi_{\text{AR}}$ on *HalfCheetah-v3* (continuous). The comparison curves are shown in Figure 11. Concretely, we have the following variants in total:

**GADM+DQN w/ extreme noise w/ E.A.R.**: The normal *GADM+DQN* agent use *Extreme Action Regression* in *GADM*, and when collecting data, we execute the random Bernoulli extreme action distribution with a small probability (e.g. 0.1).

**GADM+DQN w/ Gaussian noise w/ E.A.R.**: The variant agent use *Extreme Action Regression* in *GADM*, and when collecting data, we first add a Gaussian noise into the decoded continous action same as in (Fujimoto et al., 2018), then use the noised action interacting with the environment. Specifically, the Gaussian distribution is $\mathcal{N}(\mu, \sigma^2)$, and the clipped noise range in *[-0.5, 0.5]*.

**GADM+DQN w/ extreme noise w/o E.A.R.**: The variant agent don't use *Extreme Action Regression* in *GADM*, and when collecting data, we execute the random Bernoulli extreme action distribution with a small probability (e.g. 0.1).

Figure 11 shows that when collecting data, if we add a gaussian noise action into the decoded raw continuous action, the performance of *GADM+DQN* drops significantly. We conjecture that the reason for this is the normal action representation learning process is severely hindered by the continuous noise injection. And the performance of *GADM+DQN w/o E.A.R. + extreme noise* is better than *GADM+DQN w/ E.A.R. + extreme noise*, verifying the effectiveness of *Extreme Action Regression* discussed before.

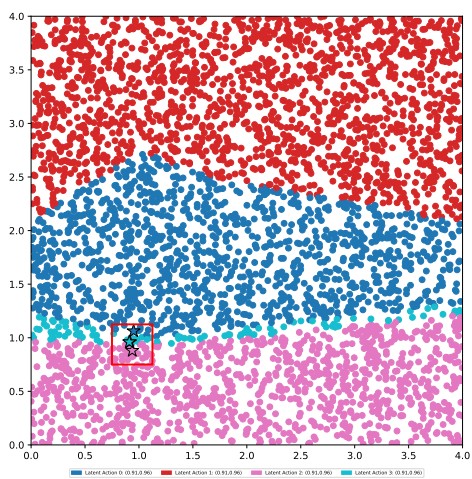 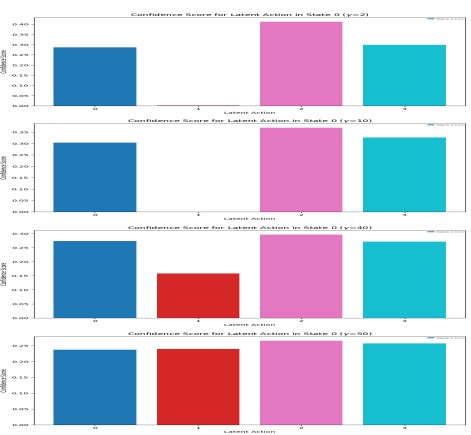

**Figure 12** Visualization of the latent action space, learned by *GADM* in ToyEnv with $K = 4$ at the specific state 1. **Left**: Mapping from the original to the latent action space. The red box indicates the In Distribution actions within ToyDataset. Colored points signify mappings to latent actions and pentagrams represent decoded actions, seen as cluster centers. **Right**: Confidence prediction for latent actions, trained with focal loss over different $\gamma$ values. As $\gamma$ increases, the confidence distribution trends toward uniformity. At $\gamma = 0$ (cross-entropy loss), confidence scores accurately mirror the latent action distribution. Note that too high $\gamma$ values lead to uniform confidence scores of $1/K$. In practice, it is found that setting 2 is a reasonable initial choice.

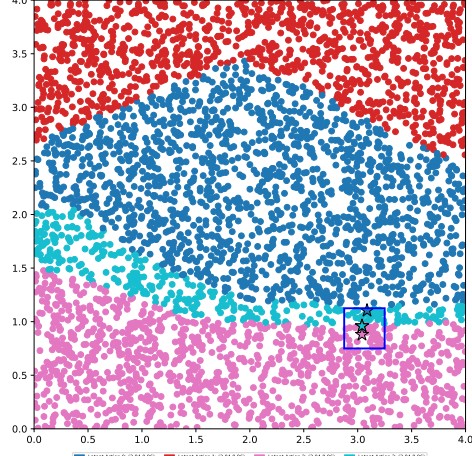 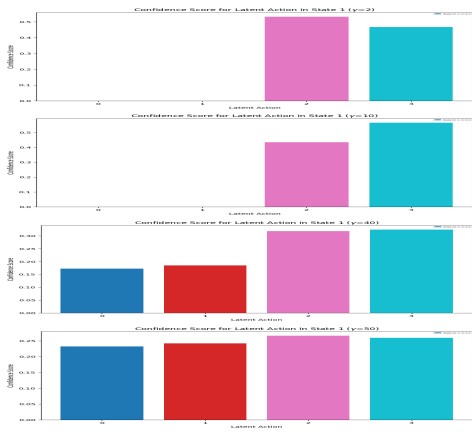

**Figure 13** Visualization of the latent action space in ToyEnv with $K = 4$ at the specific state 2.

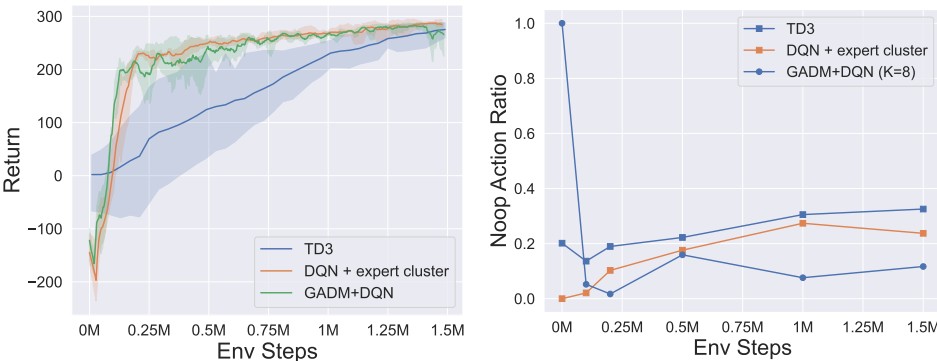

**Figure 14 left** Episode return of three algorithms on LunarLander: TD3 (raw continuous action space), DQN + expert cluster, GADM+DQN (discrete action learned by GADM from scratch) ; **right** The proportion of semantically identical actions (no operation), where a higher number of *no-op* actions denotes increased redundancy during training, hence lower is preferable.

## A.6 VISUALIZATION ANALYSIS OF LATENT ACTION SPACE

### A.6.1 OFFLINE SETTINGS

To delve deeper into the behavior of GADM in the learning process of the latent action space, we have visualized this space through a carefully constructed toy environment (ToyEnv). For ToyEnv's environment state transition, reward function, and other details pertaining to the Markov Decision Process (MDP) please refer to section .

**Setup** In an offline setting, we created the ToyEnv and its corresponding dataset, ToyDataset. The ToyEnv only includes four discrete states, with an action space that is a two-dimensional continuous space, each dimension ranging from [0,4]. To generate the ToyDataset, we uniformly sampled 4096 actions within a small square region corresponding to each state. Actions outside these regions are defined as Out-of-Distribution (OOD) actions. Next, we executed our *GADM+DQN* (K=4) algorithm on ToyDataset, conducting a visual analysis of the learned action model and the latent action confidence predictor.

**Results** As shown in Figure 12, the left graph demonstrates the mapping relationship from the original action space to the latent action space under a specific state, state0. The area marked by the red box represents the In-Distribution action area in ToyDataset. Different colored dots symbolize different latent actions mapped, while the small pentagrams represent the positions of four latent actions mapped back to the original action space via the action decoder, serving as the cluster centers for their respective categories. The right of Figure 12 illustrates the confidence prediction of different latent actions by the latent action confidence predictor, trained using focal loss with different $\gamma$ values, under the same state. The results indicate that as the $\gamma$ value increases, the confidence distribution gradually approaches a uniform distribution. When $\gamma$=0, which equates to using cross entropy loss, the confidence score of the latent actions precisely mirrors the distribution of the latent actions in the dataset. For example, under the current state0, there are no instances of latent action=1 in the dataset, which is why the predicted confidence score is also 0. However, if the $\gamma$ value is too large, all confidence scores will be $\frac{1}{K}$. A reasonable $\gamma = 2$ can mitigate the imbalance in the amount of different latent actions in the dataset without leading the optimization of confidence to approach a uniform distribution. This allows for the determination of a convenience threshold value for the confidence mask as $\frac{1}{K}$.

### A.6.2 ONLINE SETTINGS

**Case Study on *LunarLander*** To assess the performance and the proportion of non-operational actions, such as excessive *no operation*, we conduct an empirical experiment (Figure 14). Leveraging these insights, we design an action model that autonomously discerns the latent discrete action space from the raw action space. This model efficiently approximates the necessary components while discarding the redundant aspects of the raw space, thereby boosting exploration efficiency.

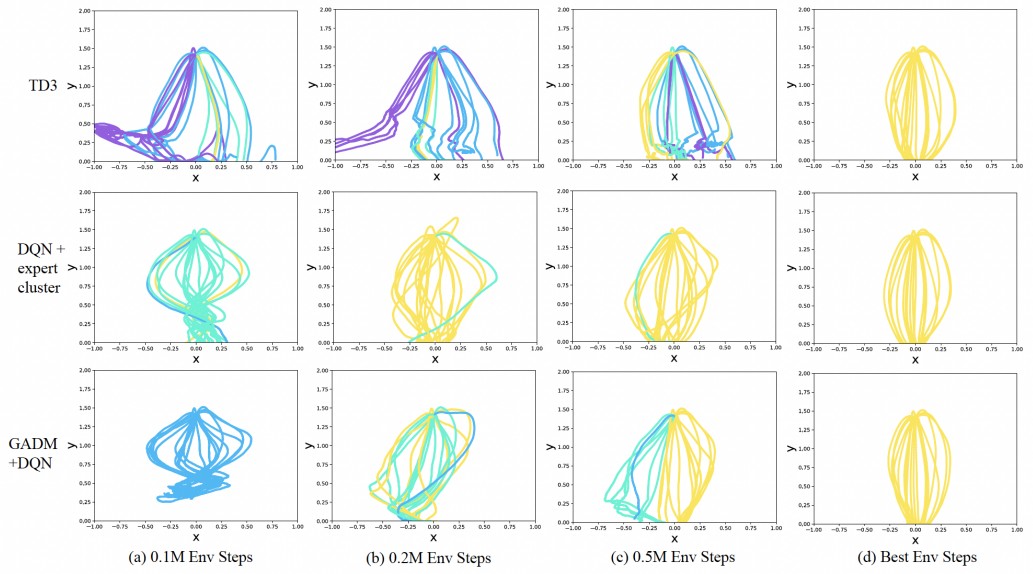

(a) 0.1M Env Steps     (b) 0.2M Env Steps     (c) 0.5M Env Steps     (d) Best Env Steps

**Figure 15** Comparison of the spaceship's landing path during different training stages of three algorithms in the *LunarLander* environment. Trajectories with color closer to red denote higher episode return, i.e., yellow > green > blue > purple, with episode return ranges of $(-inf, -100), [-100, 0), [0, 200), [200, inf)$, respectively. The algorithms compared include TD3 (raw continuous action space), DQN + expert cluster (discrete action space obtained by clustering on TD3 expert data), and GADM+DQN (discrete action learned by *GADM* from scratch).

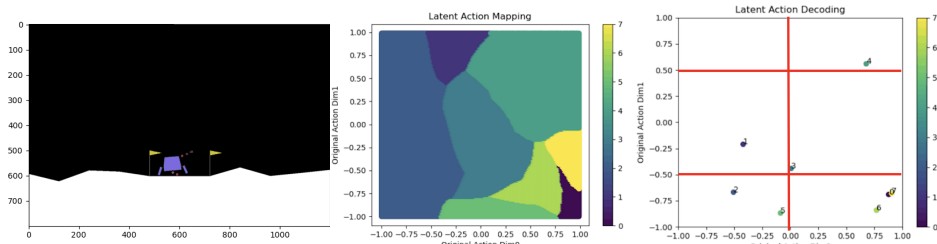

**Figure 16** Visualization of the latent action space of LunarLander games. Detailed explanation is provided in A.6.2.

Furthermore, several studies Jiang et al. (2022) have highlighted that the marginal distribution of each action dimension often exhibits multi-modal characteristics, suggesting a discrete categorical distribution might be more suitable than the simple regression used in TD3. For extended analysis, Figure 15 showcases the landing path comparison of different algorithms in the *LunarLander* environment.

Our study also illuminates the learned latent action representation in the LunarLander environment for 2-dimensional continuous control (Figure 16). Initially, the figure displays the status of the spaceship when it is about to land, i.e., launching both horizontal and vertical engines to control speed and position (left). It then uniformly samples points in the raw continuous space and transforms them with the action encoder to find their nearest discrete indexes (middle). Finally, we directly send the corresponding embeddings in the code table to the decoder to obtain their counterparts in the raw space (right). Interestingly, we observe that the learned latent space mirrors the intrinsic mechanisms of this environment, as highlighted by the red line: it uses a single discrete action to represent less important actions in this state, like *no-op*, mapping several different actions to the bottom right corner.

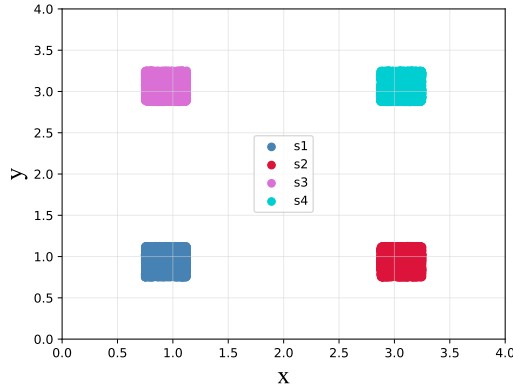

**Figure 17** Visualization of the constructed ToyDataset. The dataset is configured in a two-dimensional action space and comprises four discrete states, each corresponding to a corner of the action grid. For each state, we uniformly sample 5000 two-dimensional actions within its respective square region. During the *GADM* training, states are represented using one-hot encoding.

### A.7 DEFINITION OF TOYENV AND TOYDATASET

**ToyEnv** Inspired by (Nikulin et al., 2023), we formulated a similar environment termed as *ToyEnv*, illustrated in Figure 17. The embodied Markov Decision Process (MDP) is characterized as follows:

- *State Space:* The environment comprises four discrete states: $s1$, $s2$, $s3$, and $s4$.
- *Action Space:* The action space is a two-dimensional continuous span, denoted as $a = (x, y)$. To facilitate the subsequent construction of the offline *ToyDataset*, we define a special square action region for each state (with a side length of $0.375$). For each state, actions within this special square region are considered In-Distribution (ID) actions, and all others are Out-of-Distribution (OOD) actions. These regions are defined as follows:
  - For state $s1$, the action region is $x = [0.75, 1.125]$, $y = [0.75, 1.125]$.
  - For state $s2$, the action region is $x = [2.875, 3.25]$, $y = [0.75, 1.125]$.
  - For state $s3$, the action region is $x = [0.75, 1.125]$, $y = [2.875, 3.25]$.
  - For state $s4$, the action region is $x = [2.875, 3.25]$, $y = [2.875, 3.25]$.
- *Transitions Dynamics:* The transitions between states are driven by the $y$-value of the raw actions. For states $s1$ and $s2$, the $y$-value threshold is $0.9375$; for states $s3$ and $s4$, it is $3.0625$. The threshold corresponds to the median $y$-value within the square action region of each state:
  - If the $y$-value of a performed action is **larger** than the threshold conditioned on the current state, the system transitions to the next state, i.e., $s1 \rightarrow s2$, $s2 \rightarrow s3$, $s3 \rightarrow s4$, and $s4 \rightarrow s1$.
  - If the $y$-value of a performed action is is **smaller** than than the state-conditioned threshold, the system remains in the current state, i.e., $s1 \rightarrow s1$, $s2 \rightarrow s2$, $s3 \rightarrow s3$, and $s4 \rightarrow s4$.
- *Rewards and Termination Conditions:* Only the reward in state $s4$ is equal to $1$, and the termination condition is set to true. In all other states, the reward is $0$ and the termination condition is false.

**ToyDataset** To construct the corresponding offline *ToyDataset*, we randomly sample 5000 actions from the square region defined for each state. This approach ensures that each state is associated with a specific action region, defined by a square in terms of x and y coordinates. Through this process, we create the *ToyDataset*, a comprehensive dataset that encapsulates all possible states and its ID actions. This dataset is not only easy to understand and visualize but also provides a rigorous testing and analytical environment for the learning and exploration of our *GADM* model.

