# OpenReview forum: "Unifying Diverse Decision-Making Scenarios with Learned Discrete Actions"
_ICLR.cc/2024/Conference — Submitted to ICLR 2024_

### Official Review · Reviewer_Fy3u · 2023-10-30

**Soundness:** 3 good
**Presentation:** 2 fair
**Contribution:** 2 fair
**Rating:** 5
**Confidence:** 3

**Summary:**

Introduce a general framework that can apply common RL algorithms to a learned discrete latent actions. The authors proposed GADM, which use a modified VQ-VAE to discretize raw actions into compact latent action and predict confidence scores to help stable policy learning

**Strengths:**

In general, I think the idea to learn to compress action representation is interesting. The experiments also show interesting results.

**Weaknesses:**

1. In Figure 1, the authors show that the actions are redundant in terms of transition function or Q function. However, in the proposed method, they propose to use a modified VQ-VAE to reconstruct the action condition on the state. I think there might be a gap between the story and the proposed method.

2. The writing is sometimes confusing. It might be helpful to include necessary information in the main paper.

**Questions:**

1. In section 3.3 “diversity-aware code table”, the authors design a diversity-aware code table (“which is initialized by a series of one-hot vectors or bisection points and remains fixed through the entire training process”). As for one hot vectors and bisection points, how is the code table initialized?

From the description, it seems that the only loss is Eq.9. Compare to VQ-VAE, do you fix the code book and cancel the code book learning loss as well as the commitment loss? What is $L_d$ in Eq.9?

2. In the paragraph “latent action confidence predictor”, what does it mean by “the action model prefers to select the latent action with more recent training samples”? Does recent samples here refers to more frequently seen samples? Can you further explain why it is similar to RND?

3. Currently, evaluation seems to be limited in relatively simple / periodically locomotion tasks. As for the scalability of the method, can GADM solve relatively complicated tasks (e.g., manipulation, locomotion with obstacles)?

4. For D4RL tasks, in quite interesting to see the locomotion tasks require such a small $K$ (e.g., K=4 for hopper). Can you further explain on this? It is interesting to see what will happen if you directly discretize the action or cluster the (s,a) pairs for such tasks. This can test what is the key of the method: the proposed action model or just reducing the action size.

---

> ### Author Response · Authors · 2023-11-18
> **Rebuttal by Authors (Part I)**
>
> Thanks for your valuable feedback. We will address your comments and questions in the following parts:
> - **Question about Figure 1**: The purpose of Figure 1 is to illustrate the redundancy of action spaces, indicating that similar `(s, a)` pairs will be mapped to the same latent action. The GADM state-conditioned encoder, which we designed, precisely implements this by inputting `(s, a)` and outputting the corresponding latent action `k` (i.e., the encoded features $z^e=e_\phi(s,a)$). For more details, please refer to Section 3.2 of our revised paper.
> - **Writing**: As stated in the "General Response" section, we have made extensive revisions to our paper. Titles of sections with significant updates are highlighted in red in the latest version. We hope these modifications enhance the paper's readability and impact. We are grateful for your insightful feedback and look forward to receiving more constructive comments.
> - **Code table**:
>     - As described in Section 3.2, if we directly apply the codetable design of VQ-VAE to GADM and reuse the original exponentially weighted average method for updates, we would obtain poor results. The codetable here needs to serve as the decision-making action space for subsequent RL training. However, the original design of VQ-VAE tends to make most vectors in the codetable very similar, leading to homogeneity in decision semantics. This limits the actions that RL algorithms can select, thereby impacting final performance. A new approach is necessary to ensure the vectors in the codetable remain as diverse as possible. Since RL algorithms only need to use the index of different latent actions, it does not need to generate different styles of images like the original image generation task of VQ-VAE, we can directly fix the value of the codetable so that it does not participate in training, thereby maintaining sufficient diversity.
>    - Specific implementation methods are to initialize the codetable with a series of one-hot vectors or bisection point vectors. Suppose the code table contains $K$ candidate latent actions. The one-hot vector means that each latent action also corresponds to a K-dimensional (i.e. N=K) one-hot vector. It is equivalent to each latent action being widely separated, and the distance between two vectors are equal. The bisection point vector is to let $K$ points fall on the $K$ equal points of the line segment connected by $[-1, -1, ..., -1]$ and $[1, 1, ..., 1]$ (N-dimensional vector). Each bisection point corresponds to a vector.
> - **Latent action confidence predictor**:
>   - We have thoroughly optimized and restructured the section on the `Latent Action Confidence Predictor`. We have included an in-depth analysis of the issue concerning the pathological latent action space and have refined Figure 3. We also explain the input and output of the latent action confidence predictor, the training loss, and provide a strict definition of the latent action mask technique. We outline how the latent action mask technique can be applied in concrete discrete RL algorithms in Section 3.3 of our revised paper.
>   - Similarities to RND:
>     - The main idea of RND is to measure novelty using the prediction error in a specific state. Given the neural network's characteristics in fitting datasets for supervised learning, a larger prediction error at a certain state implies the agent has visited the nearby state space less frequently in the past, suggesting a higher novelty for that state.
>     - In contrast, the latent action confidence predictor uniformly samples `(s,k)` pairs from the replay buffer, where k is obtained by encoding `(s,a)` using the latest action encoder. We train a classifier-like model, where `(s,k)` pairs that frequently appear in the replay buffer have a stronger impact on the neural network update, thus yielding a higher confidence score. Conversely, if a pair `(s, k')` appears less frequently or not at all in the replay buffer, its impact on the neural network update is smaller, resulting in a lower confidence score. Thanks to the unique initialization of the neural network, the initial confidence score is set to $\frac{1}{K}$ (where $K$ is the total number of latent actions), so the output confidence score $C$ for a certain state reflects the distribution of `k` in the replay buffer. so the output confidence score $C$ for a certain state is proportional to the distribution of `k` in the replay buffer. While the foundational concept has certain similarities with RND, their application and detailed mechanisms show significant distinctions. Consequently, we have refined and enhanced the relevant descriptions in the latest revision of our paper.

---

> ### Author Response · Authors · 2023-11-18
> **Rebuttal by Authors (Part II)**
>
> - **Scalability**：We fully agree that testing in more complex environments would provide a better assessment of the performance of the proposed method. However, we should point out that in our MuJoCo experiments with online RL (Figure 9), we have already included quite complex environments such as Humanoid with `observation shape: 376` and  `action shape: 17`. In the hybrid environment experiments with online RL (Figure 4), the action space of HardMove-v0-n10 is also large (1024 discrete action types), and GoBigger features complex structured observations and dynamic environments. The performance in these environments has partially validated the scalability of our method. In the next revision of our paper, we will include baseline results on the D4RL antmaze environment.
> - **Analysis of the latent action shape $K$**:
>   - The good performance can be achieved with Hopper when $K=4$. We attribute this to two reasons:
>     - First, due to the structure of state-conditioned action, it's as if there are 4 actions available under each state, rather than 4 actions available under all states. This significantly expands the range of optional actions from K to $|S|^K$, where |S| is the number of all possible states.
>     - Second, according to the analysis in previous paper "Is bang-bang control all you need? solving continuous control with bernoulli policies.", decent performance can be achieved in these special MuJoCo environments in D4RL just by using bang-bang control. Therefore, the number of effective actions required in this environment is inherently limited.
>     - These two reasons explain why a small $K$ is able to obtain satisfactory results.
>   - Regarding the direct comparison of manual discretization and direct clustering, we have already carried out these experiments in the original version of our paper.
>     - You can find the simple discretization of DQN in Figure 9. As expected, due to the exponential increase in action space dimensions after discretization, the optimization of Q becomes difficult and results in poor performance.
>     - In Figure 14, you can find our case study on LunarLander environment in the online RL setup. It's observed that DQN + expert cluster, which directly uses actions from expert trajectories for clustering, then applies DQN, can achieve similar results to GADM+DQN. This indeed shows that reducing the dimensionality of the action space greatly impacts performance.
>     - However, we believe that our proposed GADM is more widely applicable, as expert data may not always be available and clustering in complex action spaces can be complicated. GADM can learn discrete actions from scratch, and in the offline RL setting, it can effectively mitigate the pathological latent action space problem (see details in the latest version of our paper, Figure 3) with our proposed latent action mask, which makes a naive DQN with GADM can draw comparable results with several spefically designed offline RL algorithms.

---

> > ### Comment · Reviewer_Fy3u · 2023-11-22
> >
> > Thanks for your response. I appreciate the endeavor you made into revising the paper. However, some of my concerns still remain: (1) I understand the Fig 1, but my question did not focus on what Fig 1 is. I wanted to ask why doing reconstruction on $a$ by given $(s,a)$ can help merging the redundancy w.r.t. transition or Q? (2) now I understand why you said it is similar to RND. But I suggest removing this part (claim your intuition is like RND) because it is not quite necessary and may cause potential confuse. (3) After reading other reviews, I agree that you should compare with other baseline. I think the action space is not a big deal if you just consider them as continuous number or cut them into bins?  Therefore, I would like to keep my score for now. But I appreciate the response and the efforts, which makes me feel the paper will be more solid in the future.

---

> ### Author Response · Authors · 2023-11-22
> **Rebuttal by Authors (Part III)**
>
> Dear Reviewer Fy3u,
>
> We greatly value your insightful feedback on our paper and appreciate the time you have dedicated to its detailed review. We are eager to address your concerns and clarify any misunderstandings.
>
> (1)
> - Indeed, Figure 1 serves as an illustrative example, emphasizing the inherent redundancy found within the raw action space. This redundancy, prevalent in transitions and $Q$-values, can be significantly mitigated through a more compact discretization process, thereby expediting the learning process.
> - Although we did not directly illustrate how this redundancy is eliminated in terms of transitions or $Q$-values in the main text, many of our experimental results indirectly substantiate this point.
>     - For instance, in the experiments conducted in the online hybrid action space, our proposed method, *GADM*, markedly outperforms existing algorithms specifically designed for hybrid action spaces. In the *Moving* environment, for example, once the action type `x` is set to `break`, the decision-making process remains unaffected by any value of the action parameter `y`. This suggests that a substantial number of raw actions are semantically consistent, enabling us to achieve performance superior to algorithms like *MPDQN*, which are expressly designed for hybrid action spaces, using our learned discretized action space.
>     - Additionally, the experiments conducted in the offline *D4RL* environment corroborate this. Even when setting a relatively small value of $K$, we managed to achieve results comparable to those of offline RL algorithms operating in the raw continuous action space.
> - We appreciate your suggestion and will delve further into how *GADM* minimizes action space redundancy from the perspectives of transitions and $Q$-values in the next revision of our paper.
>
> (2) In response to your second point, suggesting the removal of the part where we compare our intuition to that of *RND*, we concur with your perspective. Our initial intention was to offer a familiar frame of reference to help readers comprehend our method. However, we acknowledge that this comparison could potentially lead to confusion. We have already eliminated this part in the revised version of our paper.
>
> (3)
> - We agree with your suggestion, as well as those from other reviewers, that our model should be compared with more related baseline methods. However, we have some reservations about the view that "I think the action space is not a big deal if you just consider them as continuous number or cut them into bins?"
> - Indeed, for continuous action spaces, traditional methods like *TD3* and *SAC* can deliver satisfactory results. However, when dealing with complex hybrid action spaces, which are composed of continuous and discrete components, traditional RL techniques often fail to perform well. Our approach is adept at handling various decision-making problems, including hybrid and continuous ones, in both online and offline scenarios within a unified framework. This could serve as a reference baseline for future research. Specifically, the advantages of learning discrete action spaces are demonstrated through examples in hybrid action spaces.
>   - *Hybrid action space*: For instance, the action spaces of various environments tested in Figure 4. Specifically, the action space of "Moving and Sliding" is `(x, y)`, where the value of the action parameter y depends on the choice of action type `x`. In addition, the action space of the *HardMove-v0-n10* environment includes discrete actions and continuous parameters `(x, y)` — where x includes $2^{10}$ discrete actions, and y represents a ten-dimensional continuous action. In the GoBigger environment, the action space is defined as `[x, y, action_type]`. Here, `(x, y)` signifies the player's control over the ball's acceleration, while action_type is an integer representing the type of action, with `[0,1,2]` denoting three distinct types of actions. This type of action space poses significant challenges to traditional RL algorithms.
>   - In such cases, algorithms specifically designed for hybrid action spaces, such as MPDQN and HPPO, become essential. However, these algorithms are often intricately designed and lack general applicability. As shown in Figure 4, we found that *GADM+DQN* consistently outperforms baseline methods like *MPDQN* and *HPPO* in terms of both performance and stability, providing empirical support for the advantages of learning the discrete action space.

---

> ### Author Response · Authors · 2023-11-22
> **Rebuttal by Authors (Part III, Continued)**
>
> - In addition, if we choose to discretize simply, we would face the issue of exponential growth in the dimensionality of the discretized action space. As shown in Figure 9, in the experiment of the online continuous environment, the performance of *DQN (Manually Discretized)* gradually deteriorates as the action space expands. Even in some hybrid action space environments where we can manually design a fine-grained discretized action space (e.g., the GoBigger environment shown in Figure 4), our *GADM+DQN* still surpasses the *DQN w/ Manually* method to reach a higher performance ceiling.
>
> We thank you again for your thorough and constructive review.
>
> Best Regards.
>
> Authors

---

### Official Review · Reviewer_9Kqe · 2023-10-31

**Soundness:** 3 good
**Presentation:** 2 fair
**Contribution:** 3 good
**Rating:** 6
**Confidence:** 2

**Summary:**

In this paper the authors present the General Action Discretization Model (GADM), a generic approach that is applicable to any RL task that can transform their action space into a compact set of discrete units. They separate RL tasks into two modules, i.e. action model and RL model, and advocate using a separately trained VQ-VAE inspired action model, which would discretize the action space, removing redundancies, and ultimately easing the job of the RL model. The authors evaluate GADM over several online and offline RL scenarios and report promising results.

**Strengths:**

- The idea of learning a discrete action space and framing this as a generic framework for any RL task is sound.

- Paper was well written and easy to understand.

- The Background section was helpful to refresh and familiarize with the RL notation. However this can be moved to appendix and the reader could be referred to it.

- The authors conduct and share experiment results on several online and offline benchmarks, helping the reader to gauge the effectiveness of GADM better.

**Weaknesses:**

- Verbosity and the structure of the paper can improve. As is, we do not get to the actual proposed method until page 6 out of a 9 page paper. Furthermore, the appendix is longer than the paper itself (13 pages), and has some experiment results that'd be better suited for the main manuscript, such as visual samples of the learnt action samples from different scenarios. Also failure case analysis would have been nice to have.

**Questions:**

- Figure 3: It'd have been nice to have current trajectory and other collected trajectories rendered differently, as it is hard to differentiate between them.

- Why the authors use the term "codetable" instead of "code book" which is the commonly accepted VQ-VAE terminology?

---

> ### Author Response · Authors · 2023-11-18
> **Rebuttal by Authors**
>
> Dear Reviewer 9Kqe,
>
> We greatly value your insightful feedback on our paper and appreciate the time you have dedicated to its detailed review. We are eager to address your concerns and clarify any misunderstandings.
>
> - **Verbosity and Structure**:
>     - As indicated in the "General Response," we made comprehensive revisions to our paper. In the latest edition, we have color-highlighted the titles of the sections that underwent significant modifications in red. We anticipate that these changes will improve the paper's readability and influence. Please note that due to the length constraints of the main paper, the visualization analysis of the action space learned in the offline and online settings continues to be included in the appendices. However, we made clear references to it in the experiment section of the manuscript. We thank you for your insightful feedback and look forward to receiving more constructive comments from you.
>
> - **Failure Case Analysis**:
>     - The ablation experiments in both the main text and the appendix partially reflect our analysis of failure cases, e..g.:
>         - In the ablation studies for online and offline settings presented in the main body of the paper, we demonstrated that not employing our proposed techniques results in a significant performance drop. Specific methods, such as latent action mask, latent action remapping, and warmup, were extensively discussed in their respective sections.
>         - Moreover, in the appendix's experiments on online continuous action space, the performance of DQN with naive discretization was noticeably underwhelming. We have conducted an initial analysis of this. If techniques such as E.A.R. are not implemented, the performance will suffer a notable decrease, as briefly discussed in the appendix.
>      - In subsequent revisions of the paper, we plan to incorporate more comprehensive analyses of other failure cases. We appreciate your suggestion.
>
> - **Original Figure 3**:
>     - We appreciate your inspiring suggestions. In response, we have revised Figure 3 of the original paper according to your recommendations and improved the corresponding textual descriptions and captions. Please note that due to space limitations, in the revised version of the paper, we have moved the original Figure 3 to Figure 6 in the appendix.
>
> - **"Codetable" vs. "Codebook"**:
>     - We fully agree on the importance of using professional and literature-consistent proprietary terms. We referred back to the original VQVAE paper, where they use the term "embedding table," while some subsequent NLP and CV papers, such as "Zero-Shot Text-to-Image Generation", use the term "codebook". There appears to be no unified standard.
>      - Our initial choice of using "codetable" was because it reflects both a lookup table and a type of encoding. We believe that consistent use of a proprietary term in the paper with clear semantics is acceptable. If necessary, we will change it to a more suitable term in the revised version of the paper.
>
> We thank you again for your insightful feedback and look forward to receiving more constructive comments from you.

---

### Official Review · Reviewer_PWxK · 2023-10-31

**Soundness:** 2 fair
**Presentation:** 1 poor
**Contribution:** 2 fair
**Rating:** 3
**Confidence:** 5

**Summary:**

This paper proposes to discretize any action space used in RL with a state conditioned VQ-VAE using a dataset of collected experience. The authors claim this as a general solution for any action space — discrete, continuous, or hybrid.

**Strengths:**

- Formulating a general framework for various kinds of action space is an important problem in RL.
- The idea of separating action representation learning and RL is widely applicable to online and offline RL.
- The papers demonstrates the learned discrete action space can be used with different RL algorithms — DQN and MuZero.

**Weaknesses:**

The primary problem with the paper is a non-rigorous treatment of writing, prior work, and experiments.

## 1. Writing
### 1A. Incomplete Placement of the paper in the context of Prior Work
- Action representations are learned and utilized in prior work in different ways, which are not discussed or addressed:
	+ [1] separate RL into learning action representations and then performing RL over the action representations.
	+ [3, 4] learn a latent space of discrete actions during policy training by using forward or inverse models.
	+ [2, 5] propose RL frameworks that can learn with evolving or changing action spaces, and are compatible with action representations.
	+ [6] use demonstration data to extract action representations
- Alternate ways of discretizing the action space should be compared against as baselines to the proposed VQ-VAE approach:
	+ [7-9] are various approaches to discretize the continuous action space — which makes these methods applicable to hybrid action spaces as well.
### 1B. Unsubstantiated / hand-wavy claims
- "But we argue that these algorithms can naturally be seen as a special case of our framework." - How are prior works in action representation learning a special case for the proposed framework? The assumptions made are different and the kinds of representation spaces learned are different.
- "which show its potential as a general design of decision-making foundation models" - How?

### 1C. Missing Important Details
- The baselines should be introduced in writing and talked about — how they are expected to be worse in comparison to the GADM.
- Most of appendix experiments are not referenced in the main paper, so hard to find out what's relevant or not.


## 2. Approach
### 2A. Complicated design Choices skipped or not justified.
- Many important components of the method are skipped in writing or not explained / justified properly, but then referred to in the experiments section:
	+ What does "diversity-aware codetable" mean?
	+ Why is the loss weighted by the reward function in Eq. 9? This is a myopic approach to loss weighting and would make long horizon credit assignment infeasible. Also, reward norm seems like a heuristic that won't necessarily work in all environments. For instance, in negative reward environments, norm(R(s,a)) would be higher for worse actions and the action model would ignore actions that give close to zero reward — which are actually better reward values.
	+ Latent Action Confdience Predictor: How is this implemented and what is its purpose? The writing is very unclear.
	+ What is E.A.R. — it is never introduced in the approach?
	+ Warmup was never mentioned in the approach section. Anything that is important to make the method work should be discussed in the approach — for reproducibility.

## 3. Experiments
### 3A. Baselines
- For hybrid action space environments, a fairer baseline than HPPO is "Parameterized Action DDPG" because it is off-policy RL, whereas HPPO is on-policy.
- No error bars in tables — important because the performance differences are miniscule.
- GADM+DQN should be compared against standard algorithms like SAC or TD3 on continuous action space environments.
- MuZero should be compared against on some discrete action task.
- A crucial comparison should be against continuous action representation learning instead of the discretization introduced by the VQ-VAE. This is also done in prior work listed above, so it's important to show why discretization is the right way to go.

### 3B. Environments
- Experiments on standard discrete and continuous online RL environments are missing. Only hybrid action space experiments are provided, while the paper claims: "learn unified and compact discrete latent actions for different environments that even correspond to continuous or hybrid action spaces."
- Appendix: "We evaluate on 4 environments", but I only see 1 (hopper-v3). Were the results on other tasks not promising?
- Why inconsistent environments between baseline results (only hybrid action space environments) and ablations (HalfCheetah)?

### 3C. Missing Ablations
- standard VQ-GVAE v/s diversity-aware codetable
- FiLM v/s no FiLM.
- Reward normalization v/s not

### 3D. Analysis Experiments
- Sensitivity to the codebook size?

### [References]
[1] Jain, Ayush, Andrew Szot, and Joseph Lim. "Generalization to New Actions in Reinforcement Learning." International Conference on Machine Learning. PMLR, 2020.
[2] Chandak, Yash, et al. "Lifelong learning with a changing action set." Proceedings of the AAAI Conference on Artificial Intelligence. Vol. 34. No. 04. 2020.
[3] Chen, Yu, et al. "Learning action-transferable policy with action embedding." arXiv preprint arXiv:1909.02291 (2019).
[4] Kim, Hyoungseok, et al. "Emi: Exploration with mutual information." arXiv preprint arXiv:1810.01176 (2018).
[5] Jain, Ayush, et al. "Know your action set: Learning action relations for reinforcement learning." International Conference on Learning Representations. 2021.
[6] Tennenholtz, Guy, and Shie Mannor. "The natural language of actions." International Conference on Machine Learning. PMLR, 2019.
[7] Seyde, Tim, et al. "Is bang-bang control all you need? solving continuous control with bernoulli policies." Advances in Neural Information Processing Systems 34 (2021): 27209-27221.
[8] Tang, Yunhao, and Shipra Agrawal. "Discretizing continuous action space for on-policy optimization." Proceedings of the aaai conference on artificial intelligence. Vol. 34. No. 04. 2020.
[9] Metz, Luke, et al. "Discrete sequential prediction of continuous actions for deep rl." arXiv preprint arXiv:1705.05035 (2017).


Overall, looking at the experiment results shown, the two major concerns seem to be that the experiments might be cherry-picked to highlight the performant cases of GADM. Moreover, GADM seems to require a heavy amount of hyperparameter optimization, and it is not clear if when the baselines are provided with the same level of optimization, would they be generally better? If GADM is only meant to work with hybrid action space, that is fine, but then the paper should adjust its claims accordingly. A more thorough experimental evaluation is required to justify the proposed approach.

**Questions:**

- How is the extra predictor trained?
- Several questions raised above.

---

> ### Author Response · Authors · 2023-11-18
> **Rebuttal by Authors (Part I)**
>
> We greatly appreciate your insightful and thorough review. We look forward to addressing your concerns in the following sections.
>
> **1A**: We will add related works mentioned by the reviewer about action representation and action discretization in the next revised version of our paper. And we will polish the related work part to describe the connections and differences between our proposed method and previous methods more clearly.
>
> **1B**:
> - **Prior works can be regarded as a special case of our framework**: We acknowledge that this statement may be inappropriate. In our original discussion of related work, we noted that previous methods such as HyAR only learned the continuous representation of hybrid action space, while other methods only extended the discrete action representation to offline RL. In contrast, our method is designed to handle all types of action spaces and is capable of supporting both online and offline RL, which led to our initial statement. However, based on reviewer feedback, we recognize that there are indeed many other types of action representation and discretization methods. Our proposed framework cannot unify all previous methods. A more accurate description would be **"the first method to unify action representation discretization across all types of action spaces, while simultaneously supporting both online and offline RL"**. We have corrected this in our revised paper.
> - **GADM's potential as a general design of decision-making foundation models**: To design effective foundation models, drawing from the rapidly developing field of large language models can offer valuable insights. Establishing a universal training paradigm hinges on unifying input and output types. In the field of natural language processing, the model's output can be unified by predicting contextual words, where the output space is a discretized fixed dictionary. Currently, many works (e.g., BEiT v2: Masked Image Modeling with Vector-Quantized Visual Tokenizers) in the field of computer vision is exploring methods to discretize image representations. However, in regard to sequence decision-making problems and reinforcement learning, the format of the output space (i.e., action space) is highly complex. If a unified representation form can be established, and its performance is comparable or superior to classical RL algorithms, then this unity could form an important part of the decision-making foundation model. Our method is an attempt in this direction, and its effectiveness has been initially demonstrated in the environments mentioned in the paper.
>
> **1C**:
> - **Details about baselines**:
>   - We have added a brief introduction to hybrid action space algorithms in the "Online RL Results" section, and provided an analysis as to why their performance in the hybrid action space does not match up to that of GADM+DQN.
>   - In the online continuous action space MuJoCo experiments, the baseline algorithm we compared our method with is the classical TD3 method and DQN with naive manually discretized. Due to space constraints, we have only included the corresponding references and provided brief introductions to these methods.
>   - In the offline D4RL scenarios, the baseline algorithms we compared our method with include model-free methods such as BCQ, CQL, and ICQ, as well as model-based offline RL methods like DT and TT. For the comparison of our method with BCQ and ICQ, please refer to Figure 6 in the revised paper.
> - **Reference**: As mentioned in the "General Response", we have optimized and rebuilt the experimental part in the revised paper, and added accurate references for all the supplementary experiments in the appendix.
>
> **2A**:
> - **Diversity-aware codetable**: As described in Section 3.2 of the paper, if we directly use the codetable design of VQ-VAE in GADM and reuse the original embedding loss or exponential weighted average method to update it, we will obtain very poor results. This is because the codetable here needs to be used as the decision-making action space for subsequent RL training. In other words, each vector of the codetable needs to have different decision semantics as much as possible. However, the original design of VQ-VAE will make most of the vectors in codetable become very similar, that is, they may become very homogeneous in decision semantics, which will limit the actions that RL algorithms can select, thereby affecting the final performance. Therefore, a new approach is necessary to ensure the vectors in the codetable are as diverse as possible. Since RL algorithms only need to use the index of different latent actions, it does not need to generate different styles of images like the original image generation task of VQ-VAE, we can directly fix the value of the codetable so that it does not participate in training, thereby maintaining sufficient diversity. Specific implementation methods are to initialize the codetable with a series of one-hotvectors or bisection point vectors.

---

> ### Author Response · Authors · 2023-11-18
> **Rebuttal by Authors (Part II)**
>
> **2A**:
> - **Reward weighted loss function**: The key insight behind this design is that RL does not necessitate the modeling of all possible actions, which significantly reduces the learning burden of GADM. In essence, GADM does not have to accurately reconstruct all action samples, but should focus more on those actions with potential for high returns. Hence, a weighting scheme based on immediate rewards or return-to-go is necessary. However, due to our writing mistakes, the expression of this weight was not rigorous enough in the original paper. We have corrected this problem in the revised version. Specifically, for any RL environment, GADM can directly reuse the reward shaping/normalization scheme commonly used in this environment and then scale it to [0, 1] through simple transformations to serve as weights. As for the case of negative rewards mentioned by the reviewer, the ambiguity arose from our symbolic representation `norm(R(s,a))`.  The original intent was to denote a specific type of normalization method for rewards, such as $(r - r_{min})/(r_{max}-r_{min})$, or other appropriate methodologies.
> - **Latent action confidence predictor**:
>   - As stated in the "General Response", we have undertaken a comprehensive optimization and reconstruction of the section pertaining to the latent action confidence predictor, resulting in a more refined representation in Figure 3.
>     - We have incorporated a thorough analysis of the pathological latent action space problem, which is caused by redundant and shifted latent actions. This phenomenon frequently results in overestimation and subsequent instability issues affecting both offline and online RL.
>     - In addressing these challenges, we propose the development of the latent action confidence predictor. We further elaborate on the input and output specifics of this predictor, its training loss, and provide a stringent definition of the latent action mask technique.
>     - For more in-depth details, we kindly direct you to Section 3.2 in our recently revised paper.
>     - Moreover, we've outlined how the latent action mask technique can be applied in concrete discrete RL algorithms in Section 3.3 of our revised paper, titled `Practical Algorithm`. We have included this new section in the updated version of the paper.
>
> - **E.A.R.**:
>   - Extreme Action Regression (E.A.R.) is a technique specifically devised for some environments, such as MuJoCo, which require precise extreme action values as shown in paper "Is bang-bang control all you need? solving continuous control with bernoulli policies.".
>   - Acknowledging the feedback from multiple reviewers on the need for more extensive coverage of our core methods and experiments within the main body of the paper, we have made adjustments. Considering that E.A.R. is a technique tailored for specific environments rather than a universally applicable one, it is not our core contribution and we have relocated the detailed introduction of E.A.R. to the appendix under `Implementation Details` in the revised paper, while maintaining a reference to it in the main paper.
>
> - **Warmup**: We fully agree with the review's point that "any details that affect reproducibility should be included". Therefore, we have moved the motivation and principle of warmup to Section 3.1 `Dual training pipelines for GADM: offline and online RL` in the revised paper. Specifically, the GADM for online RL setting’s training pipeline can benefit from an optional warmup phase. During the warmup stage, data is accumulated via a random or expert policy, or from a pre-collected dataset. This data is used to train the action model, providing a solid foundation for the subsequent stage, as shown in Section 4.1.3.
>
> **3A**:
> - **Baseline in hybrid action space**:
>   - In the online hybrid action space environment, we selected MPDQN and HPPO as benchmarks for the following reasons:
>     - Firstly, given that our GADM+DQN is also a derivative of DQN, it is fair that we compare it with MPDQN, which stands as the top-performing algorithm that extends off-policy DQN to hybrid action spaces.
>     - Moreover, as stated in paper "HyAR", HPPO currently stands out as one of the most effective algorithm specifically designed for hybrid action spaces.
>     - While PADDPG is recognized as one of the previous algorithms for handling hybrid action spaces, its performance does not measure up to the two algorithms we have chosen for comparison.
>
> - **Error bars in tables**: In offline RL, the impact of random seeds on performance is usually not as significant as in online RL. Therefore, many previous papers, such as D4RL, BCQ, CQL, etc., only reported the average return. We adopted this setting as well. However, adding error bars can indeed help readers to better evaluate performance, and we will be including standard deviations in subsequent versions of the paper.

---

> ### Author Response · Authors · 2023-11-19
> **Rebuttal by Authors (Part III)**
>
> **3A**:
> - **GADM+DQN should be comparison with standard algorithms like TD3 on continuous action space.**：
>   - We would like to clarify that in the appendix of the original version of our paper, we have conducted a comparative analysis of GADM+DQN and TD3, as well as DQN with naive manual discretization, in the continuous action space of MuJoCo. This can be found in Figure 9 of the appendix. Our apologies if our referencing was not clear enough before, which might have caused you to overlook it. We have now added more explicit descriptions to the experiment section for easier reference.
> - **MuZero should be compared against on some discrete action task.**：
>   - Firstly, we want to clarify that the primary aim of GADM+MuZero is to demonstrate that GADM can extend existing discrete decision-making algorithms to a broader range of applications. For instance, while MuZero is only applicable to discrete action spaces, GADM+MuZero can be applied to continuous action spaces. Due to the issue of pathlogical latent actions—which we have thoroughly defined in our latest paper—there could be stability issues with the algorithm. This problem is more pronounced in offline scenarios where there is no need for interaction with the environment, prompting us to validate the principle on offline D4RL. We paid particular attention to the impact of latent action confidence prediction in GADM on instability in offline scenarios. The results were aligned with our expectations, with GADM+MuZero performing exceptionally in addressing OOD issues in offline RL.
>   - Regarding experiments in the discrete action space, we believe that when the dimension of the discrete action space is low, it is not necessary to use action representation learning methods, as the action space is already compact in such cases. For high-dimensional discrete action spaces, it might be necessary to use action representation learning methods, but this will be the focus of our future research, as our current research focuses on hybrid and continuous action spaces.
> - **Why discretization is the right way to go?**：
>   - Our framework's principal contribution lies in its initial effort to discretize all types of action spaces, while concurrently addressing the training stability issues that occur in both online and offline RL scenarios. We do not claim that the VQ-VAE scheme for action discretization is superior to other representation learning methods. However, we do recognize and highlight some insightful aspects of action space discretization:
>     - Discretization need not consider all possible action samples, but rather focuses on the most critical parts needed for the agent's decision-making process.
>     - We need to carefully discern the information that genuinely requires compression during the process of discretization. For instance, the content described in Figure 1 of our paper provides a set of examples that can guide our thinking for subsequent module design.
>     - It's essential to ensure the diversity and stability of the code table in the process of discretization.
>     - Our analysis in the motivation section, bolstered by empirical validation, suggests that discretization yields promising results, both intuitively and experimentally.
>   - Despite not being the primary focus of our current research, we understand the importance of comparison with other action representation methods. As such, we are actively exploring alternative action representation learning methods and plan to augment our research with relevant comparative experiments in the forthcoming revised paper.

---

> ### Author Response · Authors · 2023-11-19
> **Rebuttal by Authors (Part IV)**
>
> **3B**:
> - **Experiments on standard discrete and continuous online RL environments are missing**:
>   - We would like to clarify that the appendix of our original paper has already shown the comparison results of GADM+DQN, TD3, and DQN with simple manual discretization in the continuous action space of the MuJoCo environment. Currently, it is located in Figure 9 of the appendix.
>   - Regarding experiments in the online discrete action space, we believe that when the dimension of the discrete action space is low, it is not necessary to use action representation learning methods, as the action space is already compact in such cases. For high-dimensional discrete action spaces, it might be necessary to use action representation learning methods, but this will be the focus of our future research, as our current research focuses on hybrid and continuous action spaces.
> - **Appendix: "We evaluate on 4 environments", but I only see 1 (hopper-v3). Were the results on other tasks not promising?**:
>   - Similar to the previous question, our baseline experiment was conducted in four continuous action spaces in the Mujoco environment, including hopper-v3, halfcheetah-v3, ant-v3, and humanoid-v3. We apologize if the illustrations were not clear enough and may have caused some misunderstandings.
> - **Why inconsistent environments between baseline results (only hybrid action space environments) and ablations (HalfCheetah)?**:
>   - We would like to clarify that our baseline experiments were performed across eight environments. These included four hybrid action space environments and four continuous action space environments within the MuJoCo benchmark. The outcomes from these distinct sets of experiments are depicted separately in Figures 4 and 9, respectively.
>   - In the main body of our paper, we presented an ablation study (Figure 5) that was conducted on two specific environments: the continuous action space of HalfCheetah and the hybrid action space of HardMove-v0-n10. These were chosen as representative examples for their respective action space types. We believe this experiment settings provide a comprehensive view of our methodology's performance across different types of action spaces.
>
> **3C**:
> - **Standard VQ-VAE v/s diversity-aware codetable**:
>     -  As described in Section 3.2 of the paper, if we directly apply the codetable design of VQ-VAE to GADM and reutilize the original embedding loss or exponential weighted average method for updating, the results will be suboptimal. The reason lies in the nature of the codetable, which needs to serve as the decision space for subsequent RL training. Each vector in the codetable should ideally exhibit maximal decision semantic diversity. However, the initial design of VQ-VAE might lead to a high degree of homogeneity in decision semantics among the vectors in the codetable, thus limiting the action selection of the RL algorithm and impacting its final performance. To ensure the diversity of vectors in the codetable, we propose a new strategy: given that the RL algorithm only needs to use the indices of different latent actions and doesn't need to generate images of varying styles like the original task of VQ-VAE, we can keep the values of the codetable constant, ensuring its diversity during training. The specific method involves initializing the codetable with a series of one-hot vectors or bisector point vectors. We have conducted relevant comparative experiments but due to space constraints, they were not included in the paper. These results will be added to the revised version.
> - **FiLM and Reward normalization**:
>     - Our initial paper focused on the effective use of action space discretization methods in various action spaces across both online and offline settings, without including comparative experiments for FiLM and reward normalization. These were not initially included because our choice of highlighted in the "FiLM" paper, namely its ability to perform layer-wise, input-dependent transformations of feature maps. Similarly, we opted for reward normalization and weighted reconstruction loss to allocate more model capacity to the key areas within the overall action space. However, acknowledging the importance of these technical elements for the reproducibility and robustness of our findings, we've decided to incorporate comparative experiments for FiLM and reward normalization in our revised paper. Thank you for your valuable suggestion.
>
> **3D**:
> - **Codebook size**:
>     - As our latent action mask technique can partially mitigate the pathological latent action space problem (detailed analysis is provided in the revised version of the paper), we expect that the experiments on offline d4rl are not very sensitive to the codebook size, i.e., the latent action shape K. However, this needs to be confirmed by our experiments. We will add the complete experimental results and analysis to the revised version of the paper. Thank you for your valuable suggestion.

---

> ### Comment · Reviewer_PWxK · 2023-12-01
> **Appreciate the response, but need the promised experiments to validate the method.**
>
> I appreciate the improvements made in writing, but still there is a lot to be done in terms of
> - the experiments that are missing (baselines, ablations, codebook size analysis, etc)
> - missing prior work discussion
> - clarity in writing. It should be clarified what is claimed as this work's original contribution. And these contributions should be verified experimentally. "the first method to unify action representation discretization across all types of action spaces, while simultaneously supporting both online and offline RL" this by itself is not a contribution without validating why this unified scheme is useful against its alternatives.
>
> I would keep my score as this paper needs most of the experiments that were asked, and the authors agreed to.

---

### Official Review · Reviewer_3rup · 2023-11-01

**Soundness:** 3 good
**Presentation:** 2 fair
**Contribution:** 3 good
**Rating:** 5
**Confidence:** 4

**Summary:**

This paper proposes GADM, a method for learning discrete representations of action spaces. GADM applies to a variety of training settings and environment action spaces. GADM shows better results than methods that operate directly in the raw action space.

**Strengths:**

- GADM is a broadly applicable method, and the paper empirically demonstrates this. The paper shows strong GADM performance across different training settings (online and offline RL) and different environment action spaces (discrete, continuous, and hybrid). The paper compares GADM across these domains to a variety of baseline approaches.
- GADM consistently outperforms various baselines across most of the considered settings.
- Thorough experiments between the paper and supplementary.

**Weaknesses:**

- *Insufficient Method Details*: The paper doesn't sufficiently describe the method. It's not until Page 6 that the paper discusses GADM in detail. Furthermore, this discussion omits many key details about GADM and only presents them in the supplementary. For example, the main paper doesn't discuss the latent confidence prediction. It also doesn't clearly refer to the section in the supplementary that contains this content (instead referring to "other design details are also listed in Appendix A.2"). Other important details not mentioned in Sec. 3 include: using focal loss, the weight initialization of the confidence predictor, random collection warmup, and EAR.
- *Insufficient Experimental Details*: Details about the online RL results in Sec 4.1.1 are not sufficiently described. No details about MPDQN and HPPO are described anywhere in the paper. It's therefore unclear if the experiment shows a fair comparison between GADM+DQN, MPDQN and HPPO. Furthermore, the main paper never mentions the 4 environments used in Fig. 5. While the supplementary contains these details, they are crucial to describe in the main paper. Especially the details concerning the environment action space since that is what the method is learning a representation of. Details about the offline RL results are likewise lacking in details.
- *No Comparison to Action Representation Learning Methods*: As the authors stated in Sec. 5, there exists prior work that learns discrete representations of actions. For example, the paper states, "Dadashi et al. (2022) and Gu et al. (2022) propose to learn a set of plausible discrete actions from expert demonstrations to overcome the curse of dimensionality problem". Yet, as far as I can tell, GADM isn't compared to any other action representation learning methods. Instead, GADM is only compared to methods that operate directly in the raw action space. Without this comparison, I cannot assess the empirical benefit of the method over prior works.
- *Connection to Prior Work*: The connection to prior work is unclear. When referring to related work in learning action spaces, the paper states, "we argue that these algorithms can naturally be seen as a special case of our framework," but the paper doesn't explain why this is the case. The novelty of GADM and why prior works aren't applicable in the considered settings are unclear.
- *GADM Complexity*: I don't see how GADM enables "researchers to concentrate on only one of the topics" of action representation learning and RL since there are many details of GADM that are coupled with the RL process itself. GADM requires a pre-collected dataset to train the action model, strategies to mask the action space, action remapping, and handling extreme actions. All these added options affect the RL training. Furthermore, at the end of Sec. 1, the paper states the experiments demonstrate the "scalability" of the method, yet these added components seem to limit scalability.
- GADM is described as an action representation model, yet ultimately, the encoder is learning a representation of action _and state_. GADM doesn't compare to the effect of learning the action representation conditioned on the state versus not using the state information.
- $K$ is an important hyperparameter in GADM. How robust is GADM to the choice of $K$? Table 3 shows GADM performing better with a $K$ value of 4 vs. 8. How does performance vary across more values of $K$?

Generally, the presentation of the method and experiments makes it hard to assess the work properly. I recommend the authors compress Sec 3.1 and move key details from the supplementary about the method and experiments into the main paper.

**Questions:**

- What is the novelty of GADM with respect to prior work? What are the limitations of prior work that prevent them from learning latent actions that can be applied to a variety of RL settings and environments?
- Why not compare to Sampled MuZero in experimental settings from the original Sampled MuZero paper? Instead, the authors compare it in a new setting in D4RL, making it hard to verify the performance of Sampled MuZero.
- Why does the latent action dimension greatly vary between environments? For Hopper in the offline RL environments, it is 4, yet for online RL experiments, it is 256. Why do some environments require 64x more embedding table dimensions?

---

> ### Author Response · Authors · 2023-11-18
> **Rebuttal by Authors (Part I)**
>
> We highly appreciate your thoughtful feedback and thank you for the thorough review. We look forward to addressing your concerns and clarifying any misunderstandings.
>
> - **Insufficient Method Details and Experimental Details**: As stated in the "General Response", we have made extensive revisions to our paper. The titles of sections with significant updates are highlighted in red in the latest version. We hope these modifications will enhance the readability of the paper. We thank you for your insightful feedback and look forward to receiving more constructive comments from you.
> - **Connection to prior work and novelty**: We acknowledge that the statement ("other algorithms can be seen as a special case of our framework") may be inappropriate. In original related work, we noted that previous methods such as HyAR only learned the continuous part of hybrid action space. Our method, on the other hand, is capable of handling all types of action spaces and supports both online and offline RL, which led to our initial statement. However, based on reviewer feedback, we recognize that there are indeed many other types of action representation and discretization methods. Our proposed framework cannot unify all previous methods. A more accurate description would be "**the first method to unify action representation discretization across all types of action spaces, while simultaneously supporting both online and offline RL**". We have corrected this in our revised paper.
> - **Comparison to action representation learning methods**: Our method doesn't claim that action discretization by VQ-VAE scheme is better than other action representation learning methods. However, discrete action space does have many advantages, such as low redundancy, high optimization efficiency, etc. The main contribution of our approach is the first attempt to discretize all types of action spaces and simultaneously address the training stability issues derived in online and offline RL. Therefore, the experimental part is mainly to demonstrate the versatility of different types of action spaces and training settings, and the comparison with other action representation is secondary. We are also testing some action representation learning methods and are preparing to supplement relevant comparison experiments.
> - **GADM Complexity and scalability**:
>   - We have introduced a series of components that make up GADM, but it should be emphasized that most modules can be regarded as designs within GADM. The only change to the classic RL training pipeline is to use the mask obtained from the latent action confidence predictor to appropriately modify the RL loss function (such as GADM + DQN mentioned in Section 3.3). Action encoding, decoding, and remapping can be regarded as additional data pre-processing and post-processing modules, thus decoupling from the classic RL code. In contrast, for DQN, if you want to expand to the hybrid action space, the previous approach can only make major changes to the entire optimization process like MPDQN, and if you want to expand to offline RL problems, you need to add new optimization items like CQL terms, but GADM can solve all the above problems through an action representation model and generalize naive DQN to various scenarios.
>   - For scalability, we should point out that GADM can be inserted in normal discrete RL algorithms in a plug-in manner, such as passing a new `mask` variable to control the selection of target $Q$ action in DQN. And we have already validated its performance on a series of RL environments with different challenging action spaces. In our MuJoCo experiments with online RL (Figure 9), we have already included quite complex environments such as Humanoid with `obs shape: 376` and  `action shape: 17`. In the hybrid environment experiments with online RL (Figure 4), the action space of HardMove-v0-n10 is also large (1024 discrete action types), and GoBigger features complex structured observations and dynamic environments. The performance in these environments has partially supported the scalability, and we will test more environments in future version.
> - **Question about state-conditioned design**:
>   - Without the incorporation of state-conditioning, there would only be $K$ available actions across all states. However, due to the structure of state-conditioned actions, it appears as if there are $K$ available actions under each individual state, rather than $K$ actions shared across all states. This significantly expands the range of optional actions from $K$ to $|S|^K$.
>   - We are currently supplementing our experiments without state-conditioning and the preliminary results are consistent with our expectations. When state-conditioning is removed, the decrease in the number of available actions across all states leads to a significant performance decline. We will include a detailed experimental results in our revised paper. We appreciate your constructive suggestion.

---

> ### Author Response · Authors · 2023-11-18
> **Rebuttal by Authors (Part II)**
>
> - **Question about Sampled MuZero**:
>     - Firstly, we want to clarify that the primary aim of GADM+MuZero is to demonstrate that GADM can extend existing discrete decision-making algorithms to a broader range of applications. For instance, while MuZero is only applicable to discrete action spaces, GADM+MuZero can be applied to continuous action spaces. Due to the issue of pathlogical latent actions—which we have thoroughly defined in our latest paper—there could be stability issues with the algorithm. This problem is more pronounced in offline scenarios where there is no need for interaction with the environment, prompting us to validate the principle on offline D4RL. We paid particular attention to the impact of latent action confidence prediction in GADM on instability in offline scenarios. The results were aligned with our expectations, with GADM+MuZero performing exceptionally in addressing OOD issues in offline RL.
>     - Of course, we also plan to supplement our experiments with comparative studies in online settings in subsequent work.
>     - Regarding the performance issues with Sampled MuZero, there could be some performance differences since we used open-source implementation (as the original authors did not open-source their code).
>
> - **Question about latent action shape and its sensitivity**:
>     - In the online setting, we actually use a $K$ (the size of the embedding table, i.e., the latent action shape) parameter of 128, not 256; 256 represents the dimension of the embedding vector. To avoid confusion, we have optimized the naming of the variables in our hyperparameter table for clarity.
>     - In online settings, the appropriate latent action shape $K$, is often larger than what is set in offline settings.
>       - The main reason is that the action distribution in the datasets used for online settings is generally broader than that in offline datasets, thus necessitating a larger number of latent actions for modeling.
>       - In online settings, the agent interacts with the environment actively. To find the optimal trajectory, the agent needs to explore sufficiently. Theoretically, the agent will encounter a broader range of the action space. To avoid omission, we need to set $K$ larger to better model all potential actions.
>       - In offline settings, the dataset may have been collected from expert or medium policies, so the range of actions is relatively smaller, leading to a smaller $K$ setting.
>     - Our latent action mask technique can partially alleviate the pathological problem that arises in the latent action space (a detailed analysis is provided in the revised version of the paper). We expect that the sensitivity to the latent action shape $K$, will not be very high in offline or online settings. However, this needs to be confirmed through our experiments. In the revised version of the paper, we will include the complete experimental results and analysis. Thank you for your valuable suggestions.

---

> ### Author Response · Authors · 2023-11-22
> **Rebuttal by Authors (Part III)**
>
> Here, we present some supplementary ablation study results focusing on state-conditioned settings and varying K values. A more comprehensive version of these results, encompassing additional environments, will be made available in the next revised version of our paper.
>
> - In the offline D4RL *Hopper Medium-Expert* scenario, we conducted experimental comparisons of the performances under different K values for both state-conditioned and non-state-conditioned settings. As the experimental results show, in the non-state-conditioned environment, the number of available actions for all states is fixed at K, significantly constraining the expressive power of the model, leading to a significant decline in performance.
>
> | K  | 4  | 8 | 16 | 32 |
> | --- | --- | --- | --- | --- |
> | state-conditioned | 111.4 | 105.2 | 102.4 | 101.7 |
> | w/o state-conditioned | 6.5 | 17.3 | 21.4 | 17.6 |
>
> - Similarly, we conducted ablation experiments on different latent action shapes (K) under the state-conditioned setting in *Hopper Medium-Expert* scenario. In a state-conditioned setup, the number of available actions for each state becomes $|S|^K$ (where |S| is the number of all possible states), greatly enhancing the expressive power of the model. It's noteworthy that the model performs well even when K=3. However, when K is reduced to 2, considering the dataset includes both medium and expert policies, the K=2 setting struggles to effectively reconstruct all the core actions under each state, leading to a performance decline. In our tests, K=4 proved to be the optimal choice. It's worth noting that as the value of K continues to increase, although the number of optional actions for each state also increases, it simultaneously raises the difficulty of learning the Q function, resulting in a slight performance drop.
>
> | K  | 2 | 3 | 4  | 8 | 16 | 32 |
> | --- | --- | --- | --- | --- | --- | --- |
> | state-conditioned | 86.6 | 101.6 | 111.4 | 105.2 | 102.4 | 101.7 |
>
> We thank you again for your thorough and constructive review.

---

### Author Response · Authors · 2023-11-17
**General Response**

We are grateful for the valuable insights and suggestions provided by the reviewers. In response to your feedback, we have made extensive revisions to our paper. The titles of sections with significant updates are highlighted in red in the latest version:
- In Section 3, the "GADM" section: We have optimized and moved the detailed description of the *GADM* model from the appendix to Section 3. Specifically,
  - Due to space limitations, we have moved the original 3.1 `Motivation` section and Figure 3 to the appendix.
  - We have split the new Section 3.1, "Framework Overview," into two subsections: `Framework` and `Dual training pipelines for GADM: offline and online RL.` The first part gives a detailed introduction to the composition and features of the framework, while the second part describes the details of training pipelines in both offline and online settings,  analyzes potential problems in these settings, and puts forward corresponding solutions.
  - In Section 3.2, we have added a complete definition and analysis of the training loss in the `State-conditioned action encoder and decoder` section, and referenced the network structure in the appendix for readers to intuitively understand the corresponding network structure.
  - In the `Latent action confidence predictor` section of 3.2, we have added a detailed analysis of the pathological latent action space problem and have refined Figure 3. We then detailed the input and output of the latent action confidence predictor, the training loss, and a strict definition of the latent action mask technique.
  - We have added Section 3.3, `Practical Algorithms`, which elaborates on how to integrate *GADM* with general discrete decision algorithms (such as *DQN* and *MuZero*). Note that the only modification to the original discrete algorithms is the addition of a latent action mask.
- In Section 4, the "Experiment" section:
  - We have corrected the references for each section at the beginning of the corresponding questions .
  - In Section 4.1, `Main Results`, we've added necessary introductions to the benchmark environments and baseline algorithms, and adjusted the captions of the experimental graphs to facilitate better evaluation by readers.
  - We have refactored the structure of 4.2 "Abalation Results" into two parts: `The impact of latent action mask` and `The impact of latent action remapping and warmup`.
  - *E.A.R.* is a technique designed for particular environments, such as *MuJoCo*, that have specific requirements for extreme action values. We have moved the detailed introduction of *E.A.R.* to the implementation details in the appendix.
  - At the end of the section, we have added references to the appendix on the latent action shapes K and state-condition ablation, and the visualization analysis of the online and offline learning latent space.
- Due to space limitations, we have moved the details of the "Hybrid Action Space" and "Action Transformed MDP" from the original Section 2 "Background" to the appendix.
- We have corrected all known grammatical errors throughout the text and provided in-depth explanations for unclear parts.

We hope these modifications will enhance the readability and impact of the paper. And we will soon provide detailed responses addressing the concerns raised by each reviewer.

We thank you again for your insightful feedback and look forward to receiving more constructive comments from you.

---

### Meta-Review · Area_Chair_noxK · 2023-12-14

**Metareview:**

The paper proposes a method that can discretize action spaces for efficient RL training. The evaluation shows that the proposed method works better than original raw action spaces. Despite the importance of the problem domain (a general action space framework for various types of action spaces), the reviewers find several critical limitations:
- writing clarity: the writing can generally improve in terms of clarity. There are important details that are skipped (and then some later appear in the experimental section), which harm the reproducibilty. Also, there are several hand-wavyed claims (related to the potential of the method.
- experiments: as discussed by several reviewers, there are some baseline methods that could be added. The authors agreed that they are valid suggestions but the draft was never updated.
- missing comparision to prior related works and also discussion. As 3rup and PWxK mention, it is true that highly relevant related works are not compared against the proposed method (nor discussed thorouhgly).

**Justification For Why Not Higher Score:**

There are several critical weaknesses in both writing and experiments that make hard to reproduce the method and also validate the claims.

**Justification For Why Not Lower Score:**

N/A

---

### Decision · Program_Chairs · 2024-01-16

Reject